# Toward sub-second solution exchange dynamics in flow reactors for liquid-phase transmission electron microscopy

Stefan Merkens [1,2] ✉, Christopher Tollan[1], Giuseppe De Salvo[1,2], Katarzyna Bejtka [3,4], Marco Fontana [3,4], Angelica Chiodoni [3], Joscha Kruse[1,5], Maiara Aime Iriarte-Alonso[1,6], Marek Grzelczak [5,7], Andreas Seifert [1,8] & Andrey Chuvilin [1,8]

Liquid-phase transmission electron microscopy is a burgeoning experimental technique for monitoring nanoscale dynamics in a liquid environment, increasingly employing microfluidic reactors to control the composition of the sample solution. Current challenges comprise fast mass transport dynamics inside the central nanochannel of the liquid cell, typically flow cells, and reliable fixation of the specimen in the limited imaging area. In this work, we present a liquid cell concept – the diffusion cell – that satisfies these seemingly contradictory requirements by providing additional on-chip bypasses to allow high convective transport around the nanochannel in which diffusive transport predominates. Diffusion cell prototypes are developed using numerical mass transport models and fabricated on the basis of existing two-chip setups. Important hydrodynamic parameters, i.e., the total flow resistance, the flow velocity in the imaging area, and the time constants of mixing, are improved by 2-3 orders of magnitude compared to existing setups. The solution replacement dynamics achieved within seconds already match the mixing timescales of many ex-situ scenarios, and further improvements are possible. Diffusion cells can be easily integrated into existing liquid-phase transmission electron microscopy workflows, provide correlation of results with ex-situ experiments, and can create additional research directions addressing fast nanoscale processes.

Liquid-phase transmission electron microscopy (LP-TEM) is an emerging experimental technique which permits the monitoring of processes in liquid samples with nanometer-scale resolution[1,2]. LP-TEM relies on liquid cells (LCs), which are enclosures of (sub-)micrometer thin liquid layers between electron-beam transparent membranes known as windows[3]. Sophisticated LC setups have been developed through micro-electromechanical system (MEMS) fabrication techniques which can incorporate various stimuli such as electrical biasing, heating, and solution exchange, enabling in-situ and in-operando experiments[4–9].

[1]CIC nanoGUNE BRTA, Tolosa Hiribidea 76, 20018 Donostia-San Sebastián, Spain. [2]Department of Physics, Euskal Herriko Unibertsitatea (UPV/EHU), 20018 Donostia-San Sebastián, Spain. [3]Center for Sustainable Future Technologies@Polito, Istituto Italiano di Tecnologia (IIT), Via Livorno, 60, 10144 Torino, TO, Italy. [4]Department of Applied Science and Technology (DISAT), Politecnico di Torino, corso Duca degli Abruzzi 24, 10129 Torino, Italy. [5]Donostia International Physics Center (DIPC), Paseo Manuel de Lardizabal 4, 20018 Donostia-San Sebastián, Spain. [6]TECNIPESA IDENTIFICACION SL, Parque Empresarial Zuatzu, Edifício Donosti 1-3, 20018 Donostia-San Sebastián, Spain. [7]Centro de Física de Materiales CSIC-UPV/EHU, Paseo Manuel de Lardizabal 5, 20018 Donostia-San Sebastián, Spain. [8]IKERBASQUE, Basque Foundation for Science, Plaza Euskadi 5, 48009 Bilbao, Spain. ✉e-mail: s.merkens@nanogune.eu

In LP-TEM flow reactors, the enclosed fluid is connected to external reservoirs, allowing for control over its composition inside the LC. Single- and multi-inlet systems enable a broad range of flow experiments: In (electro)chemical LP-TEM experiments, fluid flow serves multiple purposes, including the renewal of reagents, removal of liquid or gaseous reaction products, including radiolytic byproducts, and the triggering of nanoscale dynamics such as nucleation, crystal growth and dissolution as well as self-assembly of nanoparticles through solution mixing/replacement[10,11]. Recent studies have also demonstrated that flow conditions significantly disrupt the radiolytic reaction network[12,13], providing additional control of the chemical environment in the electron-beam irradiated area (IA). This versatility, combined with the ability to incorporate the stimuli mentioned above[5], distinguishes flow systems from recently proposed static mixing cells[14].

The ideal design of a flow reactor depends on the application, but in general certain characteristics are required. These include a thin, electron-transparent liquid layer in the imaging area, the ability to deposit samples prior to experimentation and their immobility under flow conditions, and precise control over the liquid composition[15,16]. Additionally, eliminating gas bubbles is crucial to minimize artifacts, especially in electrochemistry experiments[17]. Moreover, for truly quantitative in-situ experiments, it is essential to achieve fluid replenishment and/or mixing dynamics in the LC that are faster than the observed process[18].

Numerous LP-TEM flow systems have been established in the scientific literature and commercialized[19–28], and the understanding of their hydrodynamic properties is increasing[29]. Based on general microfluidic considerations, solution exchange in flow reactors results from superimposed diffusive and convective mass transport[21–23]. Solute diffusion is driven by a concentration gradient and fluid flow by a pressure gradient[30]. In (sub-)micrometer-sized channels, fluid flow is typically laminar, and the velocity profile is determined by the flow resistance and the relative alignment of individual channel compartments[7,31].

According to recent studies, the Poseidon Select sample holder by Protochips Inc[21] provides convective flow rates in the IA of $\approx 10^{-5}$ m s$^{-1}$ at operating pressures of $\approx 100$ mbar[29]. These values were obtained from experimentally validated finite element simulations assuming a 150 nm thick liquid layer and the presence of a tiny bypass. Petruk et al. developed flow systems without bypass compartments leading to increased flow velocities up to $5 \cdot 10^{-2}$ m s$^{-1}$ in the IA, and thus accelerated solution renewal[24]. Such flow setups with expeditious sample renewal rates have proven to be particularly promising for special applications, e.g., pump probe imaging and diffraction; however, they come at the cost of increased operating pressure gradients ($\approx 300$ mbar were reported for 1 μm high channels)[24], eventually resulting in increased bulging and the risk of window rupture, flushing of the sample, and limited solution mixing/replacement capabilities. Various strategies were explored to counteract these effects, including microengineering to enforce window stiffness[32] and improved flow control based on pressure-driven pumping systems[16]. The Stream holder from DENSsolutions[20] appears to be a system with similar characteristics and limitations[5,6], although quantitative hydrodynamic data is not available.

Recent studies have further demonstrated that the solution replacement dynamics in the IA of available LP-TEM flow reactors are much slower than most processes studied. This complicates reliable interpretation of triggered dynamics[29]. The characteristic timescales of solution replacement for the Poseidon Select and the Poseidon 200 sample holder (both Protochips Inc[21]) were found to be limited to a few (tens) of minutes due to long feeding channels and excessive diffusion lengths depending on the specific setup and experimental conditions[29]. Similar values are expected for the Dual Flow Liquid System by Hummingbird Scientific due to geometric similarities[22]. Mølhave et al. achieved accelerated mixing dynamics and eliminated window bulging by miniaturizing the channel geometry, enabling a variety of flow and diffusion experiments and improved image quality[23]. However, a generic limitation of this recently commercialized concept (Insight Chips[19]) lays in the monolithic LC design, which restricts access to the interior of the nanochannel, thereby hindering a-priori sample deposition.

The flow systems summarized cannot combine the full range of features required for chemical flow reactors within the stated needs. As a consequence, selecting appropriate setups for LP-TEM flow experiments involves a trade-off between the multiple desired properties. In fact, diffusion has been largely overlooked as a potential major mass transport mechanism despite the advantages it can provide (i.e., slow velocity in the IA achieved at low operating pressure), presumably due to its inefficiency at millimeter scale distances. LP-TEM flow systems with a negligible contribution of convective transport in the IA have been reported but have failed to sufficiently emphasize diffusion (e.g., Poseidon 200 setup by Protochips[21] and Dual Flow Liquid System by Hummingbird Scientific[22])[11,29]. Just recently, the importance of diffusion for solution exchange, in particular replacement, was recognized by Kunnas and co-workers outlining the necessity for short diffusion paths between reservoirs of fresh solution to the IA[25].

In this manuscript, we propose a liquid cell concept, the diffusion cell, that relies on diffusive mass transport in proximity to the imaging area, tremendously improving all desirable characteristics of LP-TEM flow reactors, namely the possibility for large overall volumetric flow rates; low overall flow resistance, i.e., low pressure build-up; fast (within seconds) solution exchange dynamics with negligible flow in the IA; flexibility to mount samples prior to experimentation; and compatibility with established holders. Several designs for various applications were developed by virtual prototyping on previously validated numerical models. Physical prototypes were fabricated, and their performance verified.

## Results and discussion
### The diffusion cell concept
LP-TEM flow systems are set up by assembling MEMS-based LCs, typically flow cells, in the tip of dedicated sample holders exhibiting limitations outlined in the introduction[22]. Figure 1a illustrates a LC concept, the diffusion cell, as a means to improve the hydrodynamic properties of such systems. A micrometer-sized flow channel (height $h_{BP}$) with expected low resistance (on-chip bypass, $BP_{on}$) aims at guiding flow into the diffusion cell and around the central nanochannel (NC, width $w_{NC}$) comprising the IA in which the higher flow resistance imposes diffusion as the dominant mass transport mechanism. Note that the proposed diffusion cell design is compatible with existing LP-TEM flow sample holders as the external dimensions of the chips remain unchanged. Below, the diffusion cell concept will be tested in commercially available LP-TEM flow setups with different mass transport characteristics and compared to default flow cells ($w_{NC} = 2$ mm)[29].

### Direct flow vs. bathtub configurations
Two flow concepts have been established for LP-TEM sample holders, which differ regarding the preferential flow path through the LC[29]. In direct flow setups, most of the flow is forced into the NC inside the LC, resulting in convection-dominated mass transport and high velocities in the imaging area (usually located in the center of the LC) achieved at rather high operating pressures. In bathtub setups, micrometer-sized (off-chip) bypass channels substantially redirect the flow around the LC, and thus the NC, leading to diffusion-dominated mass transport in the IA due to very low flow velocities, and relatively low operating pressures (see also Fig. 5 in ref. 29).

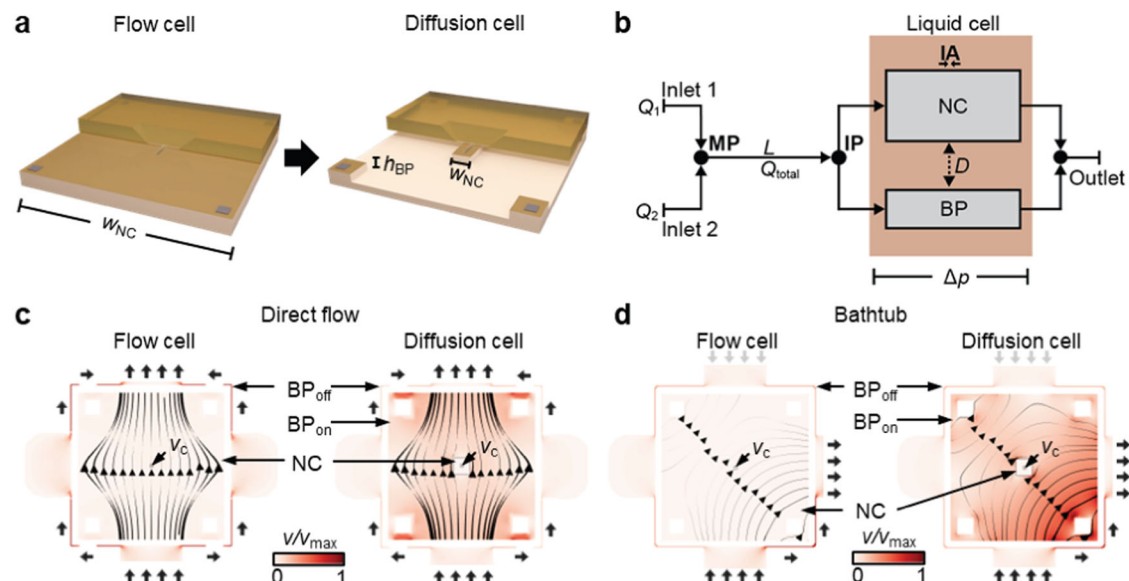

**Fig. 1 | The concept of diffusion cells for liquid-phase TEM flow systems. a** The diffusion cell concept is characterized by on-chip bypass channels (height, $h_{BP}$) that are located around a central nanochannel of drastically reduced width ($w_{NC}$) as compared to conventional flow cells constructed from two flat MEMS-based chips. **b** Schematic representation of mass transport in double inlet LP-TEM flow systems where Inlet 1, Inlet 2 – two inlet channels with flow rates $Q_1$ and $Q_2$; $L$ – distance between mixing point (MP) and the interface point (IP) - entrance into the liquid flow cell (LC); NC and BP – central nanochannel where imaging is performed (IA) and micrometer-sized channel compartment bypassing the NC (on- or/and off-chip) with corresponding flow resistances $R_{NC}$ and $R_{BP}$, respectively; $D$ – diffusion transport between BP and NC; $Q_{total}$ – total flow rate ($Q_{total} = Q_1 + Q_2$); $\Delta p$ – pressure drop developed due to the flow through NC and BP. Flow velocity profiles in flow and diffusion cells for direct flow (**c**) and bathtub (**d**) setups, respectively. NC – central nanochannel; $BP_{on}$ and $BP_{off}$ – on- and off-chip bypass channels; $v_c$ – flow velocity in the center of NC (i.e., IA). Black thin arrows (in NC) indicate direction and background color and line thickness represent magnitude of flow velocity. Short black arrows indicate active inlet (at the bottom in (**c**) and (**d**)) and outlet (at the top/right in (**c**) and (**d**), respectively) and relevant off-chip bypasses; short gray arrows (in (**d**)) indicate inactive inlets. Small black squares in the center of NC illustrate the imaging area (resulting from perpendicular assembly of windows); in the diffusion cell setup, large black squares indicate nanochannel of reduced expansion. For details on the simulations, see Methods and ref. 29.

In general, the overall pressure drop $\Delta p$ developed along either of these flow systems can be expressed by Eq. (1):

$$\Delta p = Q_{total} R_{total} = Q_{total}\left(\frac{R_{BP}R_{NC}}{R_{BP} + R_{NC}}\right), \qquad (1)$$

where $Q_{total}$ is the overall volumetric flow rate, and $R_{total}$, $R_{BP}$ and $R_{NC}$ are the flow resistance of the entire flow system, available bypass(es) and the central nanochannel, respectively.

Convective mass transport through the NC is further defined by the ratio of the flow resistances of the NC and BP (Eq. (2)):

$$Q_{NC} = \frac{R_{BP}}{R_{NC}} \cdot \frac{Q_{total}}{1 + \frac{R_{BP}}{R_{NC}}}, \qquad (2)$$

with $v_{NC} = Q_{NC}/A_{NC}$, where $Q_{NC}$ and $v_{NC}$ denote the volumetric flow rate and the mean flow velocity through the NC and $A_{NC}$ being its cross-section (Supplementary Note 1 for details on the derivation of Eqs. (1) and (2)).

**Model development – stationary flow**
Equations (1) and (2) indicate that additional on-chip bypass channels reduce the overall pressure drop (decreasing $R_{BP}$ lowers $R_{total}$) as well as the convective mass transport through the NC. To validate the beneficial effect of diffusion cells operated in direct flow and bathtub-type configurations, we performed finite-element simulations of convective transport. Figure 1c, d illustrates the flow velocity profiles through channel geometries derived from the Poseidon Select and Poseidon 200 sample holders (both Protochips Inc[21]), respectively. The width of the NC and the height of the $BP_{on}$ were $w_{NC} = 0.2$ mm and $h_{BP} = 10$ µm, respectively. In both configurations, the diffusion cell design guides flow into the on-chip bypass channel ($BP_{on}$) yet restricts

it in the central NC of reduced lateral dimensions. In particular for the direct flow configuration, the operating pressure ($\Delta p$) and the flow velocity in the center of the IA ($v_c$) are significantly decreased, from 100 to 6 mbar and from 15 to 1.5 µm s⁻¹, respectively (at $Q_{total} = 300$ µL h⁻¹; see Fig. 1c).

**Premixing vs. on-site mixing configurations**
Various characteristic locations can be defined to differentiate between multi-inlet systems as illustrated in Fig. 1b. At the interface point (IP), macroscopic capillary channels interface with the nano-channel located in the LC. The mixing point (MP) defines the merging point of (all) inflow channels. The channel compartment between the MP and the IP is usually referred to as the premixing channel with the length $L$.

Two mixing concepts have been established for LP-TEM flow reactors[22]. In premixing setups, fluids are fed through supply channels that merge far before the IP; whereas in on-site mixing setups, the supply channels reach the LC separately so that the IP and the MP coincide. By combining the presented design concepts (premixing vs. on-site mixing and direct flow vs. bathtub), 4 different (in terms of mass transport dynamics) holder configurations, all with different IP, IA and MP locations, can be envisioned. Specific implementations of such systems can comprise a negligible premixing channel ($L = 0$) and either non-existent ($1/R_{BP} = 0$) or multiple ($1/R_{BP} = \sum 1/R_i$, compare Supplementary Note 1) bypasses (BP), amongst others, without violating the generality of the description and Fig. 1b. The two most relevant sample holders studied here, the Poseidon Select and Poseidon 200 (both Protochips Inc[21]), correspond to a direct flow configuration with premixing, and a bathtub configuration with on-site mixing, respectively[29].

The flow in the diffusion cell is split between the $BP_{on}$ and the NC based on their respective flow resistances with convective mass

transport through the NC being described by Eq. (2) (see above). A solute also reaches the IA by diffusion $D$ from the $BP_{on}$.

For a general comparison of mass transport, the travel time is a suitable quantity[33]. The overall travel time, $t$, results from the respective travel times in the premixing channel, $t_L$, and in the NC, $t_{NC}$, according to $t = t_L + t_{NC}$. While $t_L$ is determined mostly by convection along $L$ (Eq. (3)), $t_{NC}$ is determined by both convection and diffusion in the NC (Eq. (4); compare Supplementary Note 1):

$$t_L = \frac{L}{(Q_{total}/A_L)} \qquad (3)$$

and

$$t_{NC} = \frac{t_D t_C}{t_D + t_C}. \qquad (4)$$

In Eqs. (3) and (4), $A_L$ is the cross-section of the premixing channel and $t_D$ and $t_C$ are characteristic times of mass transport by diffusion and convection, respectively. For simplicity, the NC is assumed to be symmetric regarding convection and diffusion (i.e., travel distance and diffusion length being both equal to half the extent of the NC, $w_{NC}/2$). $t_D$ and $t_C$ can thus be expressed as[33]:

$$t_C = \frac{\left(\frac{w_{NC}}{2}\right)}{\left(\frac{Q_{NC}}{A_{NC}}\right)} \qquad (5)$$

and

$$t_D = \frac{\left(\frac{w_{NC}}{2}\right)^2}{8D}. \qquad (6)$$

In Eq. (6), D is the diffusion coefficient of the solute in the solvent. Equations (3)–(6) point to several possibilities for decreasing $t$. First, the mixing channel can be eliminated by supplying both solutions directly on chip ($L = 0$), which sets $t_L$ to zero. This was realized in commercial holder setups, e.g., the Poseidon 200 (Protochips Inc[21]) and the Nano Channel Chip (Insight Chips[19]). Second, the volumetric flow can be increased, thereby decreasing $t_C$ following Eqs. (3) and (5). However, this is accompanied by a proportional increase in the internal LC pressure (Eq. (1)), which is intrinsically limited by the robustness of the membrane. Additionally, a high flow may influence the mobility of the sample and thus disturb the observation. Finally, both $t_C$ and $t_D$ can be reduced by reducing the lateral extension of the NC, $w_{NC}$. Note that the convective term $t_C$ depends approximately linearly on $w_{NC}$, while the diffusive term $t_D$ has a quadratic dependence on $w_{NC}$ (compare Eqs. (3) and (5) and Supplementary Note 1).

## Model development – dynamic solution replacement
We have simulated convective and diffusive transport in the above introduced models ($w_{NC} = 0.2$ mm, $h_{BP} = 10$ μm, $Q_{total} = 300$ μL h$^{-1}$) to evaluate the effect of $BP_{on}$ on solution exchange in diffusion-type LP-TEM flow reactors. The active inlet was abruptly changed from pure water to an aqueous solution of a solute (diffusion coefficient: $D = 6 \cdot 10^{-10}$ m$^2$ s$^{-1}$, concentration at inlet: $c_0 = 40$ mM)[29]. Time-resolved concentration maps of the solute in flow and diffusion cells were compared for the direct flow with premixing (Fig. 2a, b) and for the bathtub with on-site mixing (Fig. 2c, d) configuration, respectively. Diffusion-controlled solution exchange in diffusion cells completes in ≈20 s in both holder configurations (from 30 to 50 s in direct flow with premixing and 10 to 30 s in bathtub with on-site mixing configurations; compare Fig. 2e, f). The larger time delay ($\Delta t$) in the case of the premixing configuration is due to non-zero $t_L$. Interestingly, the

concentration profiles in the NC are radially symmetric in both cases (snapshots at 30 s in Fig. 2e and 10 s in Fig. 2f) indicating diffusion-controlled solution exchange. Figure 2g, h compares the time-dependent concentration profiles in the center of the NC ($\hat{=}$ IA) for direct flow with premixing and bathtub with on-site mixing setups demonstrating that the on-chip bypass leads to drastically accelerated solution exchange in both cases with respect to the default setup ($w_{NC} = 2$ mm)[29].

Based on the analysis of Figs. 1 and 2, the positive effect of diffusion cells on the hydrodynamic properties of LP-TEM flow reactors is evident. Among the multiple benefits are 1) the reduction of $t_D$ by already >1 order of magnitude, and diffusion becoming the dominant mechanism of mass transport in the NC; 2) reduction of the overall flow resistance and, consequently, reduction of the developed pressure and/or the possibility to apply significantly higher flow rates; 3) negligible convective transport in the NC, which improves sample stability; and 4) compatibility with existing LP-TEM flow holders and sample deposition protocols.

## Virtual prototyping of the diffusion cell geometry
To evaluate the potential and limitations of the proposed diffusion cell design, the influence of relevant geometrical and experimental parameters on its hydrodynamic properties was screened by simulations. Starting from parameters used for Figs. 1 and 2, simulations were performed extending the tested range for the width of the nano-channel (2 mm > $w_{NC}$ > 0.05 mm), the height of the on-chip BP channel (150 nm < $h_{BP}$ < 50 μm) and the volumetric flow rate (300 μL h$^{-1}$ < $Q_{total}$ < 3000 μL h$^{-1}$). The monitored characteristic parameters were the pressure drop ($\Delta p$) between the in- and outlet, the linear flow velocity ($v_c$) at IA, the delay ($\Delta t$) and the decay time ($\tau$) constants of solution replacement in the IA. Figure 3 summarizes the characteristic values obtained in simulations both for direct flow with premixing and bathtub with on-site mixing configurations.

A central finding of the virtual prototyping was that the lateral extension of the NC ($w_{NC}$) has no significant effect on the developed pressure (Fig. 3a) and flow velocity (Fig. 3d) in the NC (after the initial sharp drop) since the flow resistance in the NC decreases proportionally to the flow resistance of the $BP_{on}$ maintaining a balanced flow distribution. Increasing the height of the $BP_{on}$ ($h_{BP}$) decreases the pressure drop, $\Delta p$, to a negligible 0.1 mbar (Fig. 3b) and the flow velocity, $v_c$, to a few tens of nm s$^{-1}$ (Fig. 3e). This is a consequence of the drastically decreased flow resistance of the $BP_{on}$, which further allows for high overall volumetric flow rates up to 3000 μL h$^{-1}$, while maintaining pressure build-up below 100 mbar (Fig. 3c).

The achievable mixing times are characterized by $\Delta t$ and $\tau$, as introduced earlier[29] and illustrated in Fig. 2g, h. The delay time $\Delta t$ (onset of solution replacement) decreases with decreasing $w_{NC}$ reaching values as low as a few seconds for 100 μm wide NCs (Fig. 3 g). $\Delta t$ further decreases with increasing flow rate as expected from Eq. (2) (Fig. 3i). However, beyond the initial steep drop, $\Delta t$ shows no further dependence on the height of the NC (Fig. 3h). The exponential decay constant $\tau$ depends even more strongly on the lateral extension of the NC (Fig. 3j). Moreover, it remains almost constant at different $h_{BP}$ (Fig. 3k) and $Q_{total}$ (Fig. 3l), particularly in the bathtub with on-site mixing configuration (light blue curve). This unambiguously points to diffusion as the main mass transport mechanism inside the NC in both on-site mixing and premixing configurations. Even for the relatively low diffusion coefficient used in the simulations ($D = 6 \cdot 10^{-10}$ m$^2$ s$^{-1}$)[29], $\tau$ can reach ≈1 s for $w_{NC} = 100$ μm. A further decrease in NC extension does not result in a further decrease in $\tau$. This peculiar behavior was associated with intermixing of the solutions at the inactive inlet. As we will show below, improved experimental methods can further accelerate solution exchange.

Further analysis of Fig. 3 allows for in-depth understanding of the hydrodynamic properties of the flow reactor geometries (for which we

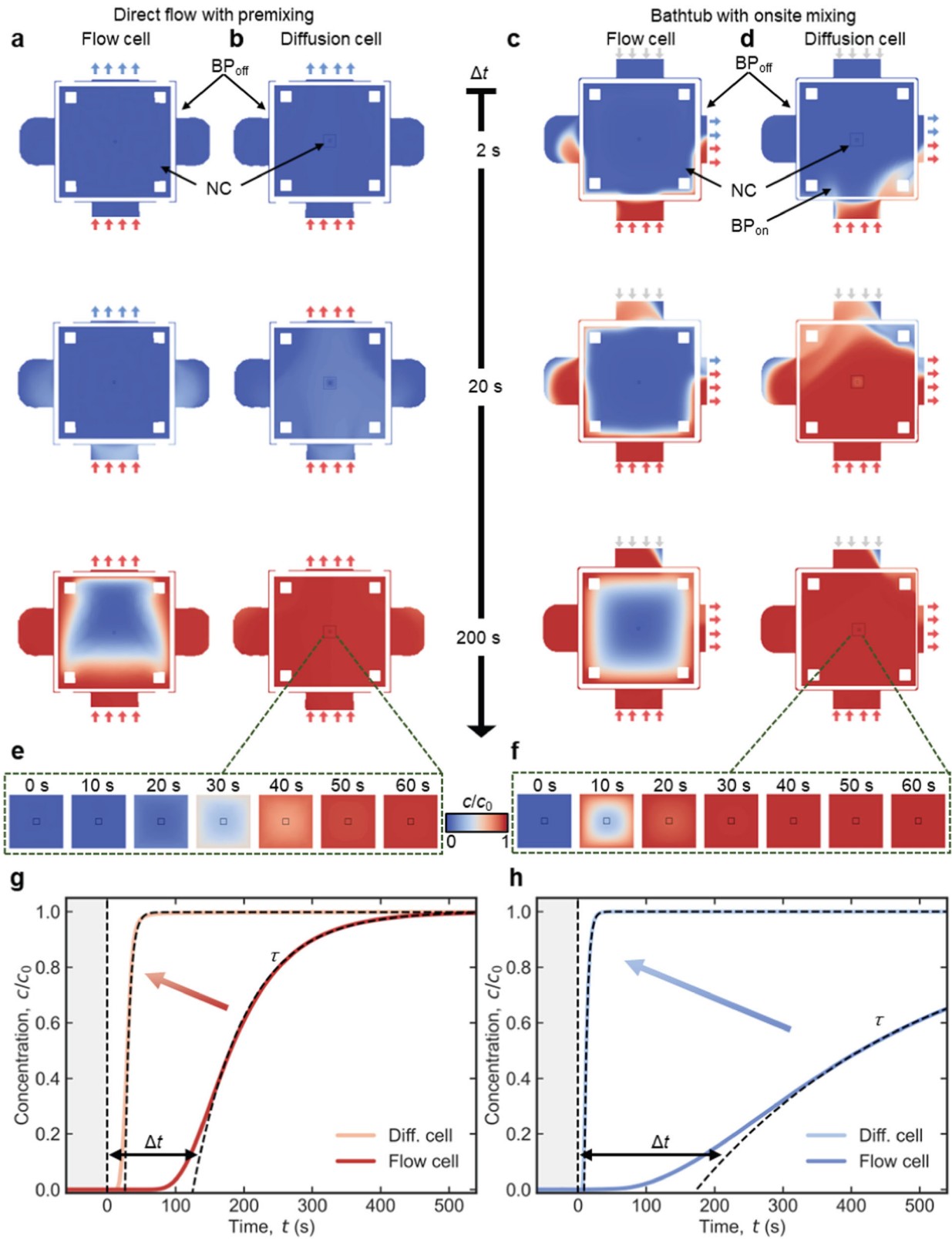

direct the reader to the Supplementary Note 3). Here, we rather emphasize the main conclusion of the virtual prototyping, namely that the inclusion of $BP_{on}$ dramatically improves the main hydrodynamic characteristic of LP-TEM flow reactors, such as mixing time (≪10 s), while allowing high total flow rates (up to 3000 μL h⁻¹), negligible convective transport in IA ($v_c < 20$ nm s⁻¹), and negligible pressure build-up (down to 0.1 mbar at $Q_{total} = 300$ μL h⁻¹).

### Device fabrication and design rules
To experimentally test the design concept, physical prototypes of diffusion cells were fabricated. For simplicity, and to demonstrate the versatility of the design, commercially available E-chips (EPB-52DNF; Protochips Inc[21]) were used[34]. Micrometer-sized bypass channels were etched into existing chips using optical lithography, wet-chemical and reactive ion etching (see Methods section for details). The

**Fig. 2 | Accelerated solution replacement in LP-TEM flow reactors operated with diffusion cells.** **a**–**d** 2D concentration maps of the direct flow with premixing (**a**, **b**) and the bathtub with on-site mixing (**c**, **d**) setup operated with default flow (**a**, **c**) and diffusion cells (**b**, **d**) at 2, 20 and 200 s after externally initiated solution replacement. Color coding represents the degree of substitution of one solution by another after abrupt switching of the flow from one inlet to the other. The flow cell and the diffusion cell form a central nanochannel with lateral expansion of $2 \times 2$ mm$^2$ (**a**, **c**) and $0.2 \times 0.2$ mm$^2$ ((**b**, **d**); indicated by large black square), respectively. The latter is surrounded by an on-chip bypass channel of 10 μm height. Small black squares in the center of NC illustrate the imaging area (resulting from perpendicular assembly of windows). Short colored arrows indicate active inlet (at the bottom in **a**–**d**) and outlet (at the top/right in (**a**, **b**) and (**c**, **d**), respectively) and qualitatively illustrate the composition of the entering/leaving solution; short gray arrows (in (**c**) and (**d**)) indicate inactive inlets. **e**, **f** Enlarged 2D concentration maps in the reduced central nanochannel (**b**) for the time range of strongest concentration variation (0 s <$\Delta t$< 60 s) for the configurations depicted in (**b**) and (**d**), respectively. **g**, **h** Time-dependent concentration curves extracted from the center of the imaging area for the four channel configurations depicted in (**a**–**d**). Black dashed lines indicate exponential fitting that allowed to extract delay and decay time constants $\Delta t$ and $\tau$, respectively. Gradient arrows in (**g**) and (**h**) indicate drastically accelerated solution replacement associated with the diffusion cell configuration. $Q_{total}$ and $D$ were 300 μL h$^{-1}$ and $6 \cdot 10^{-10}$ m$^2$ s$^{-1}$, respectively. For details on the simulation, see Methods and ref. [29].

specifications of the existing chips led to several geometric constraints (i.e., design rules) that had to be met to ensure the structural integrity of the modified chips, which resulted in the final prototypes differing slightly from the designs modeled for Figs. 1–3. The E-chips are $2 \times 2 \times 0.3$ mm$^3$ in size, have a rectangular viewing window ($550 \times 20$ μm$^2$) with an inclined groove on the back side. Flow spacers (0.2 mm × 0.2 mm × 100 nm) are located in the 4 corners. The derived chip design, which meets all requirements, is shown in Fig. 4b. The extension of the central island forming the nanochannel was measured to be ≈$120 \times 650$ μm$^2$, and the bypass channel etched on the chip ($h_{BP} \approx 10$ μm) is formed between the four pentagonal plateaus at the corners, on which the spacers are located.

## Experimental testing of diffusion cells

To test the functionality of the prototypes and confirm the predictions of the numerical model, the modified E-chips were installed in two commercially available LP-TEM sample holders with known hydrodynamic properties (see above and ref. [29]). Both direct flow with premixing- (Poseidon Select) and bathtub with on-site mixing-type (Poseidon 200; Protochips Inc[21]) configurations were tested. The orientation of the central island with respect to the flow direction was chosen so that it favors the flow, e.g., provide linear flow around the rectangular NC in the Poseidon Select setup. Simulated velocity profiles for both setups are depicted in Fig. 4a and c, respectively.

To quantify the solution exchange dynamics of the fabricated prototypes, an established contrast variation method was applied[29]. Time-dependent concentration profiles were obtained by tracking changes in the intensity of the transmitted signal as a result of alternating the flow of a highly electron-scattering contrast agent (phosphotungstic acid, PTA) and pure water. Compared to previous works, the time resolution of the experiment was significantly increased (200 ms vs. 5 s)[29]. Refer to Methods for experimental details.

The obtained time-dependent concentration curves (Fig. 4d, e) show accelerated solution replacement dynamics compared to the default flow chip ($w_{NC} = 2$ mm) by nearly two orders of magnitude[29]. Flow rates were tested in an extended range ($300 \leq Q_{total} \leq 3000$ μL h$^{-1}$). The characteristic time constants ($\Delta t$ and $\tau$; Fig. 4f–i) were extracted to obtain comparable quantitative data. The minimum delay time and decay time constant were determined to be $\Delta t = 12$ s and $\tau = 9$ s for the Poseidon Select and $\Delta t = 13$ s and $\tau = 3.7$ s for the Poseidon 200 sample holder, as compared to previous experimental results $\Delta t \approx 110$ s and $\tau \approx 120$ s and $\Delta t \approx 86$ s and $\tau \approx 156$ s, respectively[29]. For both configurations, the experimentally measured time constants were larger than the predictions of the numerical models (gray triangles in Fig. 4f–i). This discrepancy comes from the simplifications of the model, in particular neglecting the window bulging. Refer to Supplementary Note 4 for detailed discussion.

The following aspects must be considered concerning the experimental results. First, the achieved mixing times depend on the diffusion coefficient of the contrast agent, as diffusion act as the dominant mass transport mechanism. With PTA being a relatively slow diffusive species ($D_{PTA} \approx 6 \cdot 10^{-10}$ m$^2$ s$^{-1}$)[29], even faster solution exchange dynamics are expected for solutes commonly encountered in (electro-)chemical processes (e.g., sodium ions: $D_{Na+} \approx 1.3 \cdot 10^{-9}$ m$^2$ s$^{-1}$)[35], as demonstrated through developed optical methods (refer to Supplementary Note 5). Second, even though the bathtub with on-site mixing configuration can lead to faster mixing times compared to the direct flow with premixing configuration (both in simulation and experiments), the Poseidon 200 system is experimentally less reliable than the Poseidon Select system due to less control over the LC alignment in the holder tip (eventually reflected in the more pronounced deviation between simulated and modeled time constants; Fig. 4h, i)[29]. Third, the main criteria for prototyping were versatility, ease of fabrication and compatibility with existing equipment, pointing to further improvement capacities. Nonetheless, an application example described in the Supplementary Note 7 clearly demonstrates the superior capabilities for solution replacement of the existing diffusion cell setups, in particular for samples that cannot be flushed into the LC prior to the experiment.

## General guidelines

To further accelerate solution exchange, general guidelines can be concluded based on Eqs. (1)–(6) and Fig. 3, potentially leading to sub-second dynamics:

- The premixing channel should be removed because the delay time $\Delta t$ is half of the total mixing time;
- NC size should be as small as possible; for 20-μm-sized windows[34], 50 μm wide NCs seem to be the technological limit;
- the overall flow rate should be as high as possible to minimize the solution's travel time from IP to IA;
- for the same purpose, the off-chip BP should be minimized;
- to reduce the pressure drop at high overall flow rates, BP$_{on}$ should be as deep as possible; simulations show that 50 μm is sufficient for the maximum feasible flow rate.

Thus, using the terminology established above, the ideal LP-TEM mixing reactor would rely on on-site mixing (no premixing channel) and direct flow (no or minimal off-chip BP). Structurally, only Insight Chips[19] offers such systems, which however is limited to pre-assembled LCs. DENSsolutions[20] has no multi-inlet systems so far, and Protochips[21] offers either on-site mixing or direct flow, but not a combination.

## Improved experimental methodology

To challenge the limits of the diffusion cell concept, we evaluated a virtual LP-TEM configuration that meets the above criteria. The most obvious choice would be the Poseidon 200 configuration with limited off-chip BP, i.e., with integrated gasket technology. To date, such setups are not available for LP-TEM but can be accessed through simulations and replicated in ex-situ devices (refer to Supplementary Note 4 and 5 for details). Apart from the ideally designed channel geometry, sophisticated experimental methodology turned out to be even more crucial. As stated above (Fig. 3g, j), reducing $w_{NC}$ below

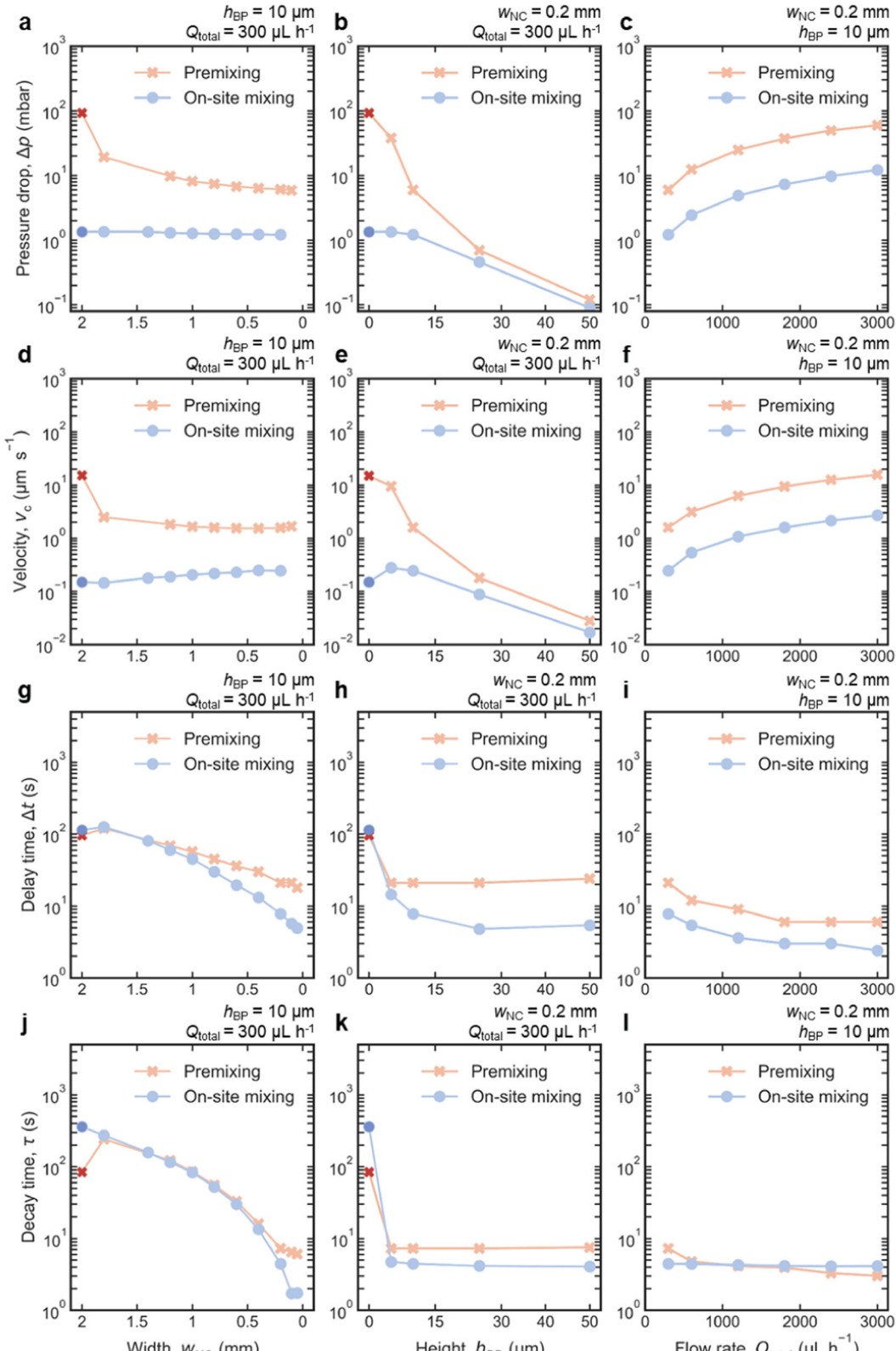

**Fig. 3 | Virtual prototyping of the diffusion cell geometry. a–l** The four characteristic hydrodynamic parameters, i.e., the overall pressure drop, $\Delta p$ (**a–c**), the velocity in the center of the imaged area, $v_c$ (**d–f**), the delay, $\Delta t$ (**g–i**), and the decay time, $\tau$ (**j–l**), constants of solution replacement, are depicted for diffusion cells operated in direct flow with premixing (light red crosses) and bathtub with on-site mixing (light blue circles) configuration, respectively. The left column (**a, d, g, j**) screens the width of the nanochannel for 2 mm > $w_{NC}$ > 0.05 mm ($h_{BP}$ = 10 μm, $Q_{total}$ = 300 μL h$^{-1}$). The central column (**b, e, h, k**) screens the height of the bypass channel for 150 nm < $h_{BP}$ < 50 μm ($w_{NC}$ = 0.2 mm, $Q_{total}$ = 300 μL h$^{-1}$). The right column (**c, f, i, l**) screens the total flow rate for 300 μL h$^{-1}$ < $Q_{total}$ < 3000 μL h$^{-1}$ ($w_{NC}$ = 0.2 mm, $h_{BP}$ = 10 μm). Dark red and dark blue crosses in the first two columns (**a, b, d, e, g, h, j, k**) represent the default flow cell ($w_{NC}$ = 2 mm; no BP$_{on}$) in direct flow with premixing and bathtub with on-site mixing, respectively[29]. The diffusion coefficient of the diffusive species was $D = 6 \cdot 10^{-10}$ m$^2$ s$^{-1}$[29]. An extended discussion of the data is provided in Supplementary Note 3.

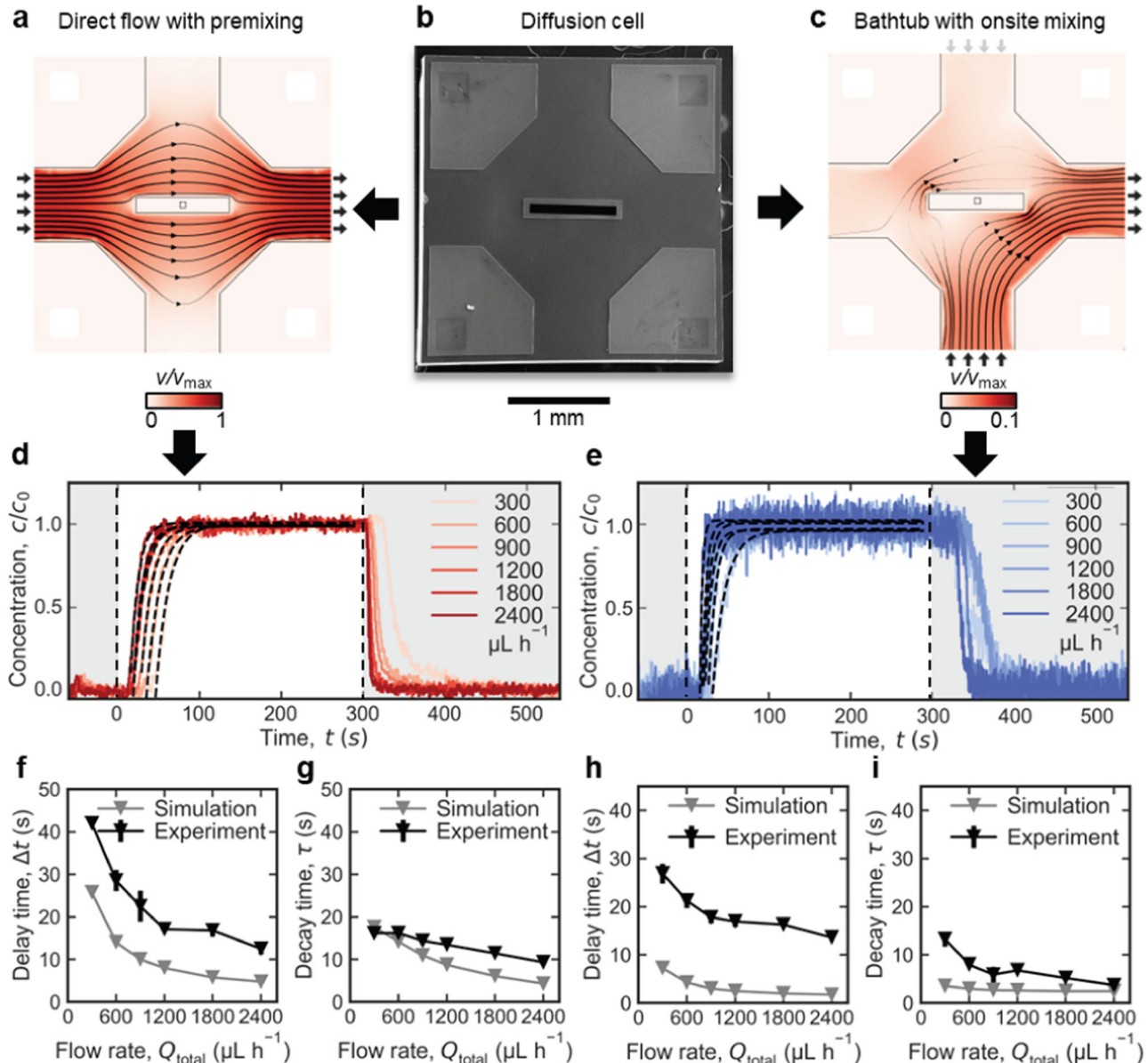

**Fig. 4 | Experimental quantification of solution replacement in direct flow with premixing and bathtub with on-site mixing LP-TEM flow reactors operated with diffusion cells. a–c** SEM image of the modified E-chip used to build diffusion cells (**b**) and simulated flow velocity fields illustrating transport through direct flow with premixing (i.e., Poseidon Select, (**a**)) and bathtub with on-site mixing (i.e., Poseidon 200, (**c**)) flow reactors, respectively. The flow is bypassing the central nanochannel where diffusion is dominant. Black thin arrows indicate direction and background color and line thickness represent magnitude of flow velocity. Short black arrows indicate active inlet (at the left/bottom in (**a**) and (**c**), respectively) and outlet (at the right in (**a**) and (**c**)); short gray arrows (in (**c**)) indicate inactive inlets. In (**a**, **c**), small black squares in the center illustrate the imaging area (resulting from perpendicular assembly of windows; compare (**c**) and the surrounding rectangles

represent the nanochannel; remaining black lines indicate channel walls of the on-chip bypass. For $Q_{total} = 300\,\mu L\,h^{-1}$, the maximum velocity inside the LC is $v_{max} = 7\cdot10^{-4}\,m\,s^{-1}$. **d**, **e** Normalized curves of contrast agent concentration (PTA; $D_{PTA} \approx 6\cdot10^{-10}\,m^2\,s^{-1})^{29}$ reflecting solution replacement in Poseidon Select (**d**) and Poseidon 200 (**e**) setups measured at flow rates $300 < Q_{total} < 2400\,\mu L\,h^{-1}$. Black dashed lines represent exponential fits. For details on the experiment and data processing, see Methods and ref. 29. Simulated and experimentally measured delay times, $\Delta t$ (**f**, **h**), and decay time constants, $\tau$ (**g**, **i**), of solution replacement obtained from exponential fitting (illustrated in (**d**) and (**e**)). Error bars in (**f**–**j**) indicate standard deviation for three independent measurements; in most cases, markers size exceeds error bars.

100 μm did not further accelerate solution replacement in the IA due to intermixing of the solutions at the inactive inlet. However, it was previously demonstrated that the inactive inlet can be assigned a negligibly small partial flow rate (PFR << 0.5) to avoid intermixing at the inlet without affecting the concentration in the NC of on-site mixing setups since the negligible flow is guided around the IA at a sufficient distance[29].

Figure 5a depicts the time-dependent concentration maps inside the diffusion cell simulated for a flow channel geometry derived from

Poseidon 200 holder featuring gasket technology, i.e., representing a direct flow with on-site mixing configuration. The relevant model parameters were lateral extension of NC: 0.05 mm × 0.65 mm; NC height: 150 nm (without considering the bulge); $h_{BP} = 50\,\mu m$; and $Q_{total} = 3000\,\mu L\,h^{-1}$ (resulting in $\Delta p = 2.5\,mbar$). Background flow was induced at the previously inactive inlet at the rate of $0.05\cdot Q_{total}$. At a steady state of $0.95\cdot Q_{total}$ water flow, a solute is mainly passing through the off-chip BP, and nothing reaches the IA (steady state, 5% flow on Fig. 5a). After enabling the solute flow at $0.95\cdot Q_{total}$, the

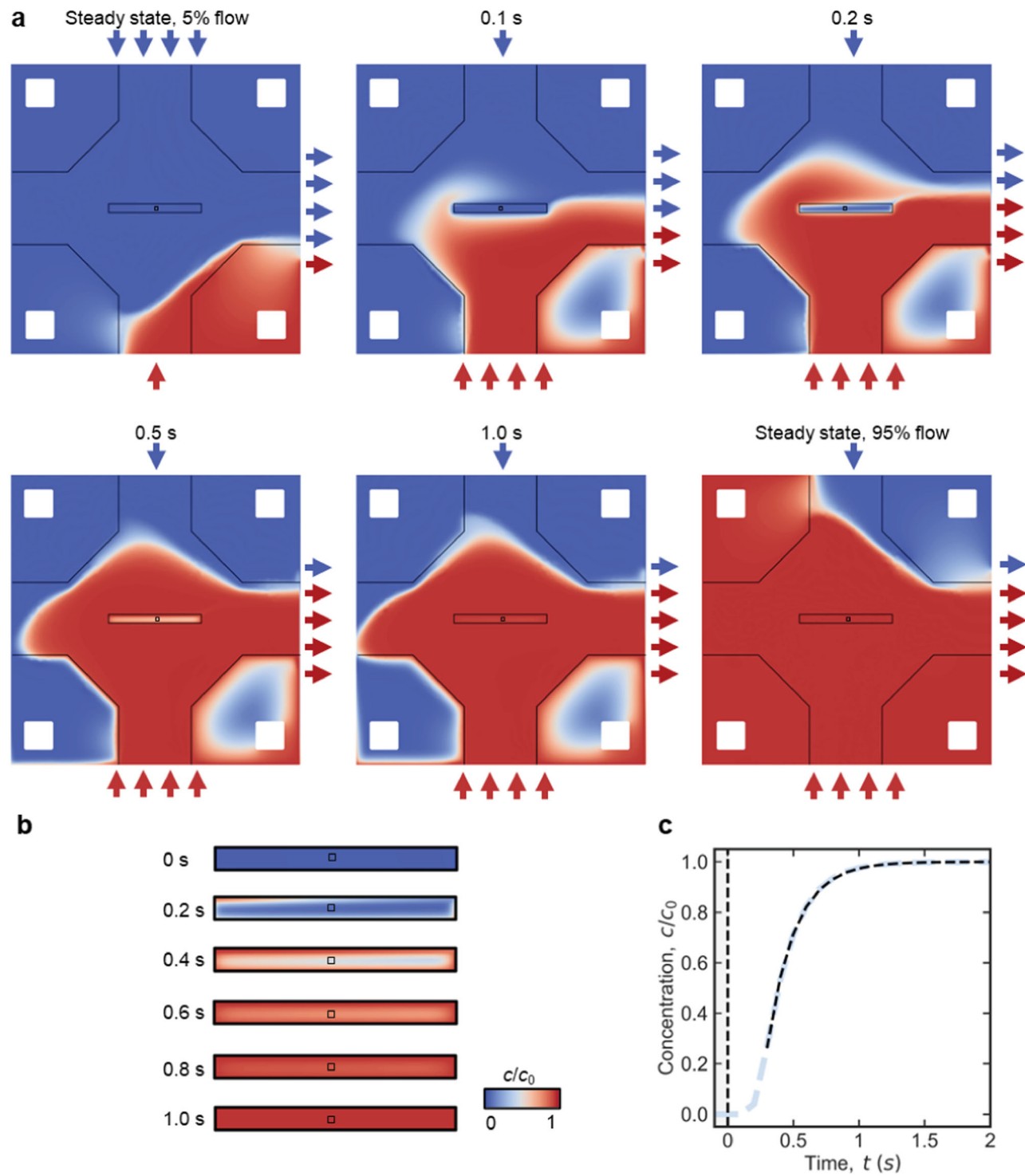

solution is completely replaced around the NC in <0.5 s. About 90% of solute replacement in the IA is reached in ≈0.7 s after the flow changeover. The characteristic time constants extracted from Fig. 5c are significantly less than 1 s, namely $\Delta t = 0.3$ s and $\tau = 0.21$ s. Note that in comparison to the above, the diffusion coefficient was adjusted to reflect more realistic scenarios of diffusing small molecules ($D = 1.3 \cdot 10^{-9}$ m² s⁻¹[35]; compare Supplementary Note 4). At steady state solute flow (steady state, 95% flow on Fig. 5a), the relative concentration in the IA ($c/c_0$) rises to 1.0 because 5% flow of pure water is confined in the nearest off-chip BP and does not reach the IA.

**Further scenarios**

Beyond the Protochips ecosystem, a virtual double-inlet diffusion cell hypothetically compatible with the Stream system by DENSsolutions[20] was created based on the geometric characteristics available for the single-inlet setup[6]. Simulated solution replacement dynamics for such devices also pointed toward sub-second mixing times (with 90% solution replacement achieved in ≈0.7 s; see Supplementary Note 6).

In addition to solution mixing, fluid renewal in the IA plays an important role in LP-TEM, in particular to remove radiolysis products and/or deliver scavengers[12]. Considering that the solution is completely renewed by diffusion in less than a second, it is in principle

**Fig. 5 | Sub-second solution replacement in virtual direct flow with on-site mixing setups operated with diffusion cells and under optimized experimental conditions. a** 2D concentration maps at different moments after externally initiated solution replacement. $t = 0$ s represents a steady state at 5% flow from the bottom inlet (solute) and 95% flow from the top inlet (pure water); at this moment the flow has been changed to 95% solute solution and 5% of pure water. Short red and blue arrows illustrate qualitatively the distribution of flow between active inlet (at the top and bottom) and the composition at the outlet (at the right). Small black squares in the center illustrate the imaging area (resulting from perpendicular assembly of windows) and the surrounding rectangles represent the nanochannel; remaining black lines indicate channel walls of the on-chip bypass. The steady state concentration maps for $Q_{CA} = 0.05 \cdot Q_{total}$ and $Q_{CA} = 0.95 \cdot Q_{total}$ are denoted as "Steady state, 5% flow" and "Steady state, 95% flow", respectively. $Q_{total}$ and $D$ were 3000 μL h⁻¹ and $1.3 \cdot 10^{-9}$ m² s⁻¹, respectively[35]. Geometric parameters were $w_{NC} = 0.05$ mm, $l_{NC} = 0.65$ mm, $h_{NC} = 150$ nm (flat − bulging neglected; refer to Supplementary Note 3 for details), $h_{BP} = 50$ μm, respectively. A gasket was assumed to block approximately 98% of the off-chip bypass. **b** 2D concentration maps of the reduced central nanochannel (**a**) are depicted enlarged for the time range of the largest concentration variation. The black square in the center corresponds to an IA of $20 \times 20$ μm² expansion. **c** Time-dependent concentration profile in IA (light blue dashed curve) induced through optimized experimental methodology, i.e., changing the actively flowing inlets between 5 and 95% of $Q_{total}$ (and vice versa). Note that despite the non-zero flow at the previously inactive inlet, the solute concentration varies between the two extrema, 0 and $c_0$ (refer to in-situ mixing discussed in ref. 29). Exponential fitting (black dashed curve) resulted in $\Delta t = 0.3$ s and $\tau = 0.21$ s. Vertical dashed black line in (**c**) indicates the timepoint at which the predominant active inlet switches from reference (pure water, gray background) to solute solution (white background).

possible to select imaging conditions in which the influence of radiolytic species would be minimized.

Another important aspect of in-situ/in-operando experiments in electrochemistry and catalysis is bubble formation in gas evolution reactions[17,36]. For example, in electrochemical LCs, gas bubbles often irreversibly block extended nanochannels and disrupt the normal conductivity of the electrolyte. The presence of a deep (tens of micrometers) on-chip BP channel with a strong fluid flow surrounding a significantly smaller nanochannel can not only prevent the formation of bubbles, but also remove already existing ones. However, the presence of electrodes imposes additional design rules for the on-chip BP geometry. In a following manuscript, we describe the development, implementation and application of diffusion cells optimized for electrochemistry[37].

In conclusion, a liquid cell concept for LP-TEM flow experiments relying on diffusion as the main mass transfer mechanism inside the nanochannel, the diffusion cell, is presented. Key advantages include 1) fast solution mixing/exchange dynamics - the fastest experimentally obtained mixing constants were ≈2 s, which is ≈2 orders of magnitude faster than previous results on default setups[29]; 2) about 2 orders reduced fluid flow velocities in the imaging region, which can be important for mechanical stability of samples; 3) about 3 orders reduced flow resistance, which positively affects window bulging and simultaneously increases the range of applicable flow rates (up to 3000 μL h⁻¹ were tested); 4) the versatility of the proposed diffusion cells allows their use with existing LP-TEM holders applying established workflows and does not limit sample preparation. A series of model experiments were presented to demonstrate the significant improvement of hydrodynamic conditions covering a broad range of applications (see also follow-up work[37]). Simulations of optimized scenarios demonstrated the ability to achieve sub-second mixing/exchange dynamics when applying sophisticated experimental methodology and pave the way for studying fast nanoscale kinetics and should allow the correlation of LP-TEM results with bulk ex-situ experiments.

## Methods
### Materials
Milli-Q water (resistivity 18.2 MΩ cm at 25 °C) was used in all experiments. PTA (99.995%) and potassium permanganate (KMnO₄, >99.0%), sodium citrate, sodium chloride, bis(p-sulfonatophenyl)-phenylphosphine (BSPP, 97%), agarose and tetrachloroauric(III) acid (99%) were purchases from Sigma-Aldrich and used without any further purification. Gasket and O-rings for sealing of the liquid cell, PTFE tubings as well as small and large Poseidon E-chips were obtained directly from Protochips Inc[21]. All used small E-chips had e-beam-transparent SiN membranes of nominal 550 μm × 20 μm expansion and flow spacers with 150 nm thickness (serial number: EPB-52DNF). As large E-chips, standard EPT-55W were used. RIE etched chips were obtained from Protochips company in the frame of a collaborative project. These

modified chips are not catalog products but can be commercially requested from the company as of now. A possible fabrication routine is described in the physical prototyping of diffusion cells section. Two different LP-TEM sample holder setups, Poseidon Select™ and Poseidon 200™ (both Protochips Inc[21]), were used for experiments. All glassware was cleaned with aqua regia and rinsed thoroughly with Milli-Q water.

### Flow reactor assembly
The assembly process of the flow reactors was identical for all setups and followed standard procedures[29]. Liquid cells, both flow and diffusion cells, were assembled in the sample holder's tip under wet conditions by enclosing tiny amounts of pristine water between two pre-treated E-chips (methanol and acetone; 10% O₂ plasma during 1 min). Standard large E-chips were combined with modified small E-chips[34]. Flow reactors were sealed using gasket and O-rings, respectively. Prior to experiments and inserting the holder into the microscope, optical, vacuum (<10⁻⁵ mbar) and flow ($Q_{total} = 3000$ μL h⁻¹ for 1 h) checks were conducted to validate the reactor assembly.

### Physical prototyping of diffusion cells
Both chemical and reactive ion etching (RIE) strategies were applied to modify E-chips. To enable chemical etching, laser sublimation was applied to write the patterns defining on-chip bypass channel in the SiN surface layer using a picosecond laser (wavelength: 355 nm; power: 0.1 W; 1 pass; line spacing: 5 μm; speed: 15 mm s⁻¹). In a second step, KOH etching (40% w/v; 80 °C) was applied to selectively etch out silicon. To obtain a depth of ≈30 μm, immersion times of ≈30 min were required. In a final step, ion beam etching (30 kV; Ga source; FEI Helios Nanolab 600) was applied to remove overhanging material and smooth the edges, especially of the central nanochannel. RIE etched chips were obtained from Protochips company in the frame of a collaborative project relying on standard deep RIE (DRIE, Bosch etch)[38].

### Image contrast variation method
The procedure for hydrodynamic quantification of flow reactors was adopted from the method of (image) contrast variation described in ref. 29. In brief, the flow is alternated between an electronically dense contrast agent solution (40 mM PTA) and a reference solution (Milli-Q water) to alter the transmitted intensity over time. In LP-TEM experiments, the (normalized) transmitted intensity, $I_{norm}$, can be expressed based on Eq. (7):

$$I_{norm} = \frac{I}{I_0} = \approx 1 - \rho_{CA} \cdot z \cdot c = 1 - K \cdot c, \tag{7}$$

where $I_0$ and $I$ denote the transmitted intensity through a LC containing pure water and contrast agent solution, respectively. $z$ is the thickness of liquid layer and $\rho_{CA}$ as well as $c$ are scattering power and concentration of the contrast agent, respectively.

## Optical contrast variation method

For ex-situ experiments, the image contrast variation method was adjusted for optical spectroscopy. In optical experiments, $I_{norm}$ is obtained from the absorbance, $A$:

$$A = \varepsilon z c \qquad (8)$$

$$A = \log_{10}\left(\frac{1}{I_{norm}}\right) \qquad (9)$$

Equation (8) is known as Beer-Lambert law and commonly used to determine the concentration of solutes (e.g., optical dyes) absorbing light in the UV-vis regime, with $\varepsilon = \varepsilon(\lambda)$ denoting their wavelength-dependent molar extinction coefficient. The wavelength of maximal extinction ($\lambda_{max}$) was determined by acquiring $A$ in the spectral range from 200 to 800 nm in ex-situ UV-vis measurements (liquid thickness $z_{liquid} = 1$ cm; compare Supplementary Note 5).

As chemical aspects (e.g., dimerization effects) limit the linearity of Beer-Lambert law to rather low solute concentrations, most optical dyes are ruled out for flow experiments unless their high extinction coefficients found in literature. $KMnO_4$ ($\varepsilon_{max} = 2.66 \cdot 10^4 M^{-1}cm^{-1}$ at $\lambda_{max} = 546$ nm)[39] was identified as suited candidate. Saturated solutions of $\approx 7$ mM were used.

UV-vis measurements were performed on a Cary UV-Vis-NIR 3500 spectrometer (Agilent Technologies Inc). The window of the liquid cell was aligned to be fully located inside the beam path (spot size: 1 mm diameter). Average absorbance ($A$) curves from the entire window area were acquired at an interval of 0.1 s between data points.

## Flow control

Two electronically connected programmable syringe pumps (Pump 11 Pico Plus Elite, Harvard Apparatus, USA) were used to control liquid flow through the two-inlets reactor. One of the syringes contained pure demineralized water (Millli-Q; resistivity: 18.2 MΩ cm at 25 °C), the other an aqueous solution of contrast agents at concentrations to ensure ≥10% decrease of transmitted intensity of pure water solution – that was, 40 mM for PTA in LP-TEM flow experiments[29]. The electronically interfaced pumps were programmed to provide a constant volumetric total flow rate ($Q_{total}$), yet abruptly alternating the applied flow between the reference (water; $Q_{water}$) and contrast agent ($Q_{CA}$) solution at constant time intervals (usually every 5 min) to study the dynamics of solution exchange. $Q_{total}$ was studied in an elevated range from 300 up to 3000 μL h$^{-1}$, by the minimum reasonable flow rate and maximum rate safe for the window membranes (representing a 10× increase in respect to ref. 29.

## Data acquisition and post-processing

TEM imaging was performed on a Tecnai G2 F20 S-twin microscope (FEI), operated at 200 kV. The same imaging conditions were used for all the experiments: parallel beam illumination, 2250× magnification, beam intensity adjusted to a dose rate of approximately 8 e$^-$ nm$^{-2}$ s$^{-1}$ (without the sample). Local image intensity ($I$) curves were averaged from a $1 \times 1$ μm$^2$ area in the center of the viewing window and acquired on a Gatan Orius 2 K*2 K CCD camera (GATAN, USA) at an interval of 0.2 s between data points.

Data acquisition was manually synchronized with the pump's microcontroller with an accuracy <1 s. The typical duration of the experiments was 10 min (1 datapoint each 200 ms). Raw intensity curves were loaded in customized Python-2.7 workflows (numpy, scipy, matplotlib) for further semi-automatized processing and plotting[40].

## Convection diffusion model

An experimentally validated numeric model for convection and diffusion in realistic 3D LP-TEM flow reactor geometries was adopted from ref. 29 for virtual prototyping. The details of the model implementation (i.e., underlying physics, geometric model refinement, meshing, definition of boundary conditions) and its validation for Poseidon 200 and Poseidon Select type sample holders (Protochips Inc)[21] were described in ref. 29. In brief, the laminar flow module was used to solve the incompressible Navier-Stokes equation coupled with the equations for Fick's laws describing water as an incompressible fluid and the diffusion of solvated species therein. For the boundary conditions, all channel walls including membranes were considered chemically inert and rigid (membrane bulging was considered qualitatively for model refinement); no liquid slip along and no penetration through the walls was permitted; inlets were transparent, and inflow was described by the volumetric flow rate; outlets were transparent and described by zero pressure. Stationary as well as time-dependent solutions were calculated to obtain stationary velocity and pressure maps as well as local time-dependent solute concentration profiles, respectively.

Due to the highly parametrized implementation of the initial model[29], virtual prototyping was feasible through reasonable screening of model input parameters. Most relevant parameters were the lateral extension of the central nanochannel, $w_{NC}$, (defining the width of the on-chip bypass channel), the height of the on-chip bypass channel, $h_{BP}$, the diffusion coefficient $D$ of the solute as well as the total volumetric flow rate, $Q_{total}$. Screening of geometric parameters was performed within the intrinsic limitations of the setup, i.e., dimension of the small E-chip (2 mm × 2 mm × 0.2 mm), with $w_{NC}$ and $h_{BP}$ ranging from 0.05 to 2 mm and 150 nm to 50 μm, respectively. $Q_{total}$ was screened in the range 300 μL h$^{-1} \leq Q_{total} \leq 3000$ μL h$^{-1}$. Diffusion coefficients were selected in agreement with experiment (where appropriate) and ranged between $10^{-9}$ m$^2$ s$^{-1}$ and $10^{-11}$ m$^2$ s$^{-1}$. For PTA, $D$ was assumed as $D_{PTA} = 6 \cdot 10^{-10}$ m$^2$ s$^{-1}$[29].

## Gold nanoparticle (AuNP) synthesis and characterization

AuNPs were synthesized using the reversed Turkevich method for citrate-capped AuNPs[41]. Functionalization of AuNPs with BSPP was done according to ref. 42. In brief, AuNPs were stirred in a BSPP solution (0.5 mL, 78.4 mM, 25 °C) overnight, then centrifuged twice (6000 × g, 120 min) and washed with Milli-Q water for storage.

## Agarose film preparation and characterization

Agarose solutions were prepared adapting previously established routines[42]. To the aqueous solution containing AuNPs@BSPP (7.5 μL, 25 mM), Milli-Q water (32.5 μL), hot agarose (400 μL, 1% w/v, 90 °C) and NaCl (40 μL, 1 M) solutions were added consecutively.

To obtain agarose thin-films, 2 μL droplets of the prepared solution were spin-coated on unmodified large E-chips after plasma activation. The spinning duration was set to 15 s and the film thickness was controlled by adjusting the spin speed between 20 and 100 revolutions per second (rev s$^{-1}$). Atomic Force Microscopy (AFM, Agilent 5500 AFM; Keysight, Santa Clara, USA) was performed under 100% humidity to characterize thickness of the hydrated thin films. To this end, manual scratching tests were performed as reported elsewhere[43,44] by employing a sharp steel tweezer to remove the gel and applying a constant force within a selected region. The uncovering and the identification of the support served to provide a defined area to further measure the height of steps of the samples. Images of 512 × 512 pixels were acquired in hydrated samples in contact mode in Milli-Q water. A DNPS silicon nitride probe <10 nm nominal radius, 0.12 N m$^{-1}$ force constant, and a resonant frequency of 34 kHz (Bruker, Madison, USA) was used for scanning. Prior to each measurement, the probe was cleaned with acetone and absolute ethanol. During the scanning, the sample was always covered with Milli-Q water and the set point was continuously adjusted to minimize the applied force. The images were examined by using picoView 1.14 software (Keysight, Santa Clara, USA).

## LP-TEM imaging of nanoscale dynamics

LP-TEM experiments were prepared by spin coating 1% w/v agarose gel with nominal thickness of 150 nm on a big chip (70 rev s$^{-1}$, 15 s; compare above) matching the selected spacer height (see below for details). Together with modified small chips, a diffusion cell was assembled in the tip of a bathtub with on-site mixing-type sample holder (Poseidon 200, Protochips Inc). The relevant geometric parameters of the flow setup comprised the height of the on-chip bypass (10 μm) and the extension of the central nanochannel (120 × 650 μm$^2$) rendering diffusion the dominant mass transport mechanism in the nanochannel.

TEM imaging at a magnification of 27000× was applied to image particle movement in changing liquid environment with a temporal resolution of 2 s (acquisition time of 1 s per image and dose rate of 5 e$^-$ nm$^{-2}$ s$^{-1}$). During the 8 min of experiment, the flow of aqueous NaCl ($c = 100$ mM) solution was interrupted by pure water for 5 min starting 1 min after the image acquisition was initiated. Supplementary Movie 1 depicts the obtained image sequence at a rate of 30 frames per second. The sequence was analyzed after the experiment by tracking the motion of particles using ImageJ software (TrackMate plugin[45]).

## Data availability

The data that support the findings of this study are available from the corresponding author upon request.

## Code availability

Custom codes used for this study are available from the corresponding author upon request.

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

## Acknowledgements

This work was supported by the Basque Government (PIBA 2018-34, RIS3 2018222034, KK-2023/00001) and Diputacion Foral de Gipuzkoa (RED2018, RED2019). We acknowledge support by Spanish MINECO under the Maria de Maeztu Units of Excellence Program (MDM-2016-0618). S.M. acknowledges funding from the Basque Ministry of Education in the frame of the "Programa Predoctoral de Formación de Personal Investigador no Doctor" (grant reference: PRE_2019_1_0239). This work has received funding from the Piedmont region (Italy) through the SATURNO project (POR FESR funding 2014–2020). This study was carried out within the Ministerial Decree no. 1062/2021 and received funding from the FSE REACT-EU - PON Ricerca e Innovazione 2014–2020. Special thanks are extended to the staff of Protochips Inc company for the ongoing technical support and in particular to Madeline Dukes and John Damiano for fruitful discussions and motivation of this work. The authors acknowledge the contribution of Alexander Bittner for characterizing spin-coated agarose gel films.

## Author contributions

S.M. – diffusion cell design idea (equal), conceptualization (equal), quantification experiments (lead), gel sample preparation and characterization (supporting), application experiment (lead), simulations (lead), data curation and analysis (lead), writing original draft (lead), review and editing (supporting); C.T. – manufacturing of physical prototypes (lead), review and editing (supporting); G.D. – simulations (lead), data curation and analysis (lead), review and editing (supporting); K.B. – quantification experiments (supporting), data curation and analysis (lead), review and editing (supporting); M.F. – quantification experiments (supporting), data curation and analysis (lead), review and editing (supporting); A.Chi. – quantification experiments (supporting), data curation and analysis (lead), review and editing (supporting), resources (equal); J.K. – gel sample preparation and characterization (supporting), application experiment (supporting), review and editing (supporting); M.I. – gel sample preparation and characterization (lead), review and editing (supporting); M.G. – quantification experiments (supporting), gel sample preparation and characterization (supporting), application experiment (supporting), data curation and analysis (lead), review and editing (supporting), resources (equal); A.S. – review and editing (supporting), resources (equal); A.Chu. – diffusion cell design idea (equal), conceptualization (equal), supervision (lead), data curation and analysis (supporting), writing original draft (lead), review and editing (supporting), resources (equal), funding (lead).

## Competing interests

The authors declare no competing interest.
