## [Peer Review File · Nature Communications]

Toward sub-second solution exchange dynamics in flow reactors for liquid-phase transmission electron microscopyREVIEWER COMMENTS

Reviewer #1 (Remarks to the Author):

Please see my comments in the attached pdf file.

Review Article: “Towards Sub-second Solution Exchange Dynamics in Liquid-Phase TEM Flow Reactors” by Merkens et al.

Reviewer Comments:

The authors have put forth a technically robust manuscript that employs both simulations and experiments to introduce a new nanofluidic liquid cell (nLC) design concept aimed at enhancing mass transport and liquid sample exchange. Termed “diffusion cells”, this nLC design offers a novel approach to conducting liquid-phase electron microscopy (LPEM) experiments requiring liquid mixing. The proposed design builds upon modifications to existing nLC chips commercially available from two major LPEM OEMs. It’s commendable that this technically sound work has successfully navigated the editorial process, as such contributions often go overlooked. I therefore recommend this manuscript for publication in a journal of the caliber of Nature Communications. However, there are specific concerns that require the authors’ attention.

The authors should revise their introduction and claims to avoid inadvertently conveying a misleading impression to the scientific community. Specifically, the work needs to be contextualized appropriately in light of existing developments in flow LPEM systems.

To elaborate, the authors must address the importance of reducing window bulging in flow LPEM systems by connecting the outlet tube to an air-evacuated container. This adjustment will set the background pressure in the outlet of the nLC at approximately $P_{\text{outlet}} = 30$ mbar, assuming the use of pure water for simplicity. The assumption made in the simulations that $P_{\text{outlet}} = 0$ raises questions about its actual value in the conducted flow experiments.

The authors seem concerned about the $\Delta P = 300$ mbar calculated in [Petruk et al., Nanotechnology 2019], a value that is substantially lower than the background pressure level of 1000 mbar – the typical value experienced by nLC in most experiments that were performed with flow LPEM systems with the outlet tube open to the lab atmosphere.

The manuscript appears to make an apples-to-oranges comparison by suggesting that diffusion can be more effective than convective mixing. Forced mixing should inherently be more efficient than relying solely on diffusion. The authors should clarify this potential misconception, particularly considering that the ‘true’ flow LPEM system presented by Petruk et al. offers sample refreshing rates about 1000 times higher than their proposed method.

The pressure differential between inlet and outlet for liquid to transverse a rectangular channel can be calculated using the following expression:

$$\Delta P_{\text{inlet-outlet}} = \frac{12 \rho \nu ECL}{CW S P t^3}$$

where $\rho = 1 \text{ g cm}^{-3}$ is the density of water, $\nu = 0.892 \text{ mm}^2 \text{ s}^{-1}$ is the kinematic viscosity of water at 298K, and V is the volumetric flow rate. The remaining parameters define the channel geometry; CW is the channel width, $S P t$ is the channel height, and ECL is the effective channel length. This expression can be rewritten as a function the refreshing time t as follows,

$$\Delta P_{inlet-outlet} = \frac{12 \rho \nu ECL^2}{S P t^2 t}$$

As the authors could see, increasing t from 1 ms (Petruk et al.) to 1 s would decrease $\Delta P_{inlet-outlet}$ a thousand times and make it rather negligible when ECL is in the order of 50 – 100 μm .

Therefore, the authors should properly compare their design concept with those with ‘true’ flow systems. The expressions provided below are in good agreement with results from COMSOL Multiphysics.

In this regard, the authors should also reference early groundbreaking work conducted by the teams of Bau [Grogan and Bau 2010 *J Microelectromech. Syst.* 19 885–94] and Miller [Mueller C et al. 2013 *J. Phys. Chem. Lett.* 4 2339–47], which, to the best of my knowledge, were developed independently of the offerings from the primary LPEM OEMs.

Lastly, the label “flow cell state of the art” in Figure 1a is misleading. This figure actually depicts a “conventional” nLC, consisting of two “flat” chips separated by a spacer. From an engineering perspective, these conventional nLCs are not configured to support true flow dynamics. Instead, as the authors themselves have pointed out, they function as diffusion cells. Their design, which features a large w_{NC} , limits their efficiency in mass transfer and liquid flow mixing. These traditional nLCs prioritize cost-efficiency, aiming to minimize the number of nanofabrication steps required for production.

In contrast, the authors propose a straightforward yet effective solution. Their design adds only a single additional RIE step to the manufacturing process. However, it's important to note that the feasibility of incorporating this modification by existing LPEM OEMs hinges on whether it would infringe upon any existing intellectual property rights.

Additional minor comments:

“(…) simulations were performed extending the tested range for the width of the nanochannel ($2 \text{ mm} < w_{NC} < 0.05 \text{ mm}$)”. If I understand correctly, w_{NC} was varied $2 \text{ mm} > w_{NC} > 0.05 \text{ mm}$.

Reference 31 appears to have an incorrect link. Please elaborate regarding the technological limit for 30 μm windows. How much does this limit depend on the material and thickness?

Please clarify the difference in the flow path between the bathtub / onsite mixing setup shown in **Figure 2** versus the bathtub / onsite mixing setup shown in **Figure 4**. It appears that there are multiple inlets in the setup shown in **Figure 2** but only one inlet in the setup shown in **Figure 4**. The inactive inlet is discussed and illustrated in **Figure 5** but perhaps this can be clarified in figures which appear earlier in the text as well.

Reviewer #2 (Remarks to the Author):

Reviewer #3 (Remarks to the Author):

Here in this article, the authors have proposed a novel design of TEM liquid cells that functions on the principle of diffusive mass transport, where they have fabricated the design based on their results from numerical simulations and have characterized the working regime.

The primary mixing time points and related discussions significantly depend on the relationships derived for the mixing time (and as mentioned on Page 7), and we notice here that the relationship derived is actually less absolute than the "mixing time"-claims outlined in the manuscript would suggest. For example, in equation S12.13, the units aren't satisfied. Also, the authors need to better describe the need to generalize the equation by eliminating terms such as viscosity, which deviates them from the authors' heavy claims to be the fastest mixing TEM flow cell.

For these reasons, the authors are suggested to significantly tone down their claims (that sound very absolute), especially not to mention that they have developed a flow cell with the fastest/shortest mixing time when they have performed the simulations and experimentations only to a limited scope of sample regime (i.e., viscosity, the inclusion of particles, particle types, device dimensions, etc.,). Under certain real-world conditions, the quoted mixing times would be much longer than the claims in the paper imply.

In Figure 3 a, d, g, j, the x-axis is plotted in descending order while all other plots in the comparison are in ascending order. This change can be confusing to the reader. Further, to take into account of plots from Fig. 3, and its corresponding claims of mixing times, the authors are suggested to add more data points (especially at short mixing times where there are too few) to observe a smooth transition/curve through the data.

In Line 344, the authors have mentioned the inclined groove. The authors are suggested to provide detailed dimensions of the groove since it is important for the TEM measurement considerations.

In the experimental testing of diffusion cells, the authors have briefly mentioned the bulging of the nanochannels. Since the mixing time period and its claims therein in the entire publication depend on these dimensions, and as mixing by diffusion is directly proportional to the channel dimensions, it appears necessary that the authors perform further experimental and numerical analyses on the same. With respect to numerical analysis, for bulging, it would be expected to perform time-resolved studies that include fluid-structure interaction to accurately model the channel shape during bulging, and not just truncated shapes (that currently seem arbitrary) as described by the authors in Fig. S5. For experimental analysis, bulging of channels can be characterized at various flow conditions, such as flow rate (& fluid conditions such as viscosity, particle density, etc.) using tomographic or detailed spectroscopic analysis. Maybe there are also options to accurately map out the bulging deformation at different relevant conditions using the TEM signal. In the hydrodynamic characterization using spectroscopy, the authors have performed the analysis at 10 Hz, while discussing about such short timestamps, the authors are encouraged to collect data at a higher frequency. In Fig. S7b, the authors have provided limited results, and more data points would be required. Overall, The authors have only briefly touched upon the bulging using spectroscopic analysis, further analysis needs to be performed on both experimental and numerical analyses. In Fig. S9, d, and e, it is clearly observed that there is a significant deviation between experimental and numerical analysis, thus in line with the previous comments. Hence, the authors are encouraged to further investigate their numerical analysis and reevaluate their experimental results in Fig. S9, S10, and S11 in the supplementary and in the main

draft Fig. 4 d, e, f, g, h, i.

In the fabrication of ex-situ Sample holders (6.1), the authors have mentioned about the Poseidon Select sample holder as the primary-based holder for further fabrication and assembly, and thereon for numerical, analytical, and experimental analyses. Hence, it is highly recommended to provide the patent number of the device and its dimensions of it. If not provided, it would go against the requirements of replication of the results by other users based on this paper. Also, it would be necessary to provide the direct flow/mixing onsite mixing setup.

In nanochannel mixing studies, it is quite common to investigate the clogging parameters by studying using various particle densities, shapes, and sizes, but no such experimentation, numerical analysis, and discussion have been performed in this article.

In the methods section, in Line 548, the authors have mentioned that both chemical and reactive ion etching strategies are utilized, while the details of reactive ion etching aren't described, nor the patent number has been provided (in case of IP protection). The authors are encouraged to provide a detailed fabrication procedure of the chip since the paper revolves around the development of a new liquid cell for TEM. Without more details, the value for the readers would be severely impacted due to the difficulty of and time needed for replicating the devices or results.

In the convection-diffusion model, the authors have assumed zero pressure at the outlet, which raises a major query and integrity of the entire numerical analysis performed in this manuscript, as obtained zero pressure is nearly impossible in such a TEM flow setup, and considering zero pressure creates higher momentum thus resulting in shorter mixing time and reduced bulging, which is quite not the case.

Also, the authors are encouraged to provide why data points have been collected experimentally every 0.2 s (Line 584) without any scientific conclusion, which is very close to the authors' claims, thus raising concern about the experimental data integrity.

On a minor note, there is a small typo in Line 606. The authors are encouraged to provide details on how the sample was prepared, how the diffusion coefficients were selected, and their references.

In this article, the authors have obtained at a single point in the nanochannel to get the mixing ratio, which doesn't include the distribution within the entire window. Thus, it would be necessary to provide a detailed analysis through the entire window spatially and temporally, where the authors would be encouraged to provide an analysis of the flow range, particle distribution, particle density, viscosity, and mixing ratios. Also, in order for this device to be considered in the future by other users, it would be expected to provide universal relationships and a database to determine the mixing time points and corresponding fluid and flow conditions.

In conclusion, this manuscript demonstrates an interesting development in the liquid cell TEM field, but it needs major revision with respect to experimental results, numerical analysis, analytical derivation, and explanations prior to further considerations.

Reviewer #4 (Remarks to the Author):

Reviewer #5 (Remarks to the Author):

The authors report on a novel liquid flow cell architecture for two liquid flow holder systems from Protochips utilized in liquid-phase TEM. Therefore, they developed a suitable layout by flow simulations, which is in a second step fabricated and tested in operational conditions. Their concept strongly reduces the flow resistance by bringing the channel inlet closer to the viewing area. This is achieved by modification of liquid cell chips by reactive ion etching. The novel concept is shown to strongly reduce solution exchange times while maintaining a purely diffusive solution exchange within the viewing area. In my opinion, the idea is brilliant and should be published after a minor revision as it solves some of the most important issues in LP-TEM that are so far strongly limiting its performance, repeatability, and versatility.

- In the modified LC, bare silicon (plus a native oxide layer) is exposed to the sample solution. This should have an influence on the stability and, thus, substances compatible to the chip. I don't expect this to be a problem as the structuring using photolithography and reactive ion etching can easily be integrated into an optimized fabrication process in which silicon is structured before silicon nitride deposition. Nevertheless, for the prototype version, a comment should be added into the SI section. Furthermore, in the methods section I think there is a RIE step for SiN removal missing between photolithography (this is only the structuring of the photoresist and not of the subjacent layer) and bulk micromachining.
- The authors mention that removal of radiolytic products is strongly enhanced due to the high exchange rates. As this is a very important aspect from an experimental point of view, there should be some more discussion. How does the novel architecture influence gas bubble formation (e.g. H₂), i.e. to what extent is a bubble formation delayed in comparison to conventional setups. I assume this can be estimated by comparing the generation of H₂ at a given dose rate with the removal rate by diffusion at a given flow rate up to a critical supersaturation value at which a bubble formation is expected.
- How does the new architecture influence the redox chemistry (i.e. solvated electrons, H and OH and HO₂ radicals)? Is it comparable to conventional setups, or does it facilitate removal of reactive species?
- In the same regard, to what extent is the acid-base chemistry (i.e. H⁺ and OH⁻) influenced?
- The authors state that the reduced flow resistance positively affects window bulging as less pressure is expected to reduce the outward bulging of the LC membranes. As window bulging is neglected in their simulations, two questions remain: Firstly, how does the complete chip bulge after assembly? In the shown configuration, spacers of 150 nm are in the four edges of the chip whereas the nanochannel has no spacer. Depending on the metal lid holding the chips in place, I would expect an outward bulging of the whole chips due to the remaining positive pressure separating both chips and probably determining the true diffusion channel height to be larger than the spacer thickness. Secondly, is the window bulging simulated in SI section 5.2. comparable to experimental values?

Reviewer #6 (Remarks to the Author):

The work by Merkens et al. presents an advanced silicon nitride liquid cell design that improves mixing during liquid phase transmission electron microscopy experiments. While the results are important and noteworthy for the LP-TEM community, they are more appropriate for a journal that specializes on microscopy. The work would become more significant in the field, and appropriate for Nature Communications, by utilizing the improved cells to study a problem that was not practical to study before with LP-TEM. The conclusions of improved flow are supported by the data analysis, and there are no substantial flaws in analysis, interpretation, or conclusions. The methodology is sound, but there is minimal experimental data in the manuscript compared to other manuscripts in Nature Communications. There is not enough detail to reproduce the manuscript as indicated by the lack of RIE information in the fabrication procedure.

REVIEWER COMMENTS

Reviewer #1 (Remarks to the Author):

The authors have put forth a technically robust manuscript that employs both simulations and experiments to introduce a new nanofluidic liquid cell (nLC) design concept aimed at enhancing mass transport and liquid sample exchange. Termed “diffusion cells”, this nLC design offers a novel approach to conducting liquid-phase electron microscopy (LPEM) experiments requiring liquid mixing. The proposed design builds upon modifications to existing nLC chips commercially available from two major LPEM OEMs. It’s commendable that this technically sound work has successfully navigated the editorial process, as such contributions often go overlooked. I therefore recommend this manuscript for publication in a journal of the caliber of Nature Communications. However, there are specific concerns that require the authors’ attention.

We acknowledge the positive evaluation and constructive feedback of the reviewer. Below we will address the reviewer’s concerns point-by-point and are convinced that this will resolve all confusions.

The authors should revise their introduction and claims to avoid inadvertently conveying a misleading impression to the scientific community. Specifically, the work needs to be contextualized appropriately in light of existing developments in flow LPEM systems.

We are grateful for the critical evaluation of the introduction by the reviewer. In that part of the manuscript, we have tried to give a comprehensive, yet concise, overview of LP-TEM systems that have been developed in the recent past, emphasizing double-inlet setups because of their ability for solution replacement/mixing. Based on the reviewer’s feedback, we have revised the introduction.

In particular, based on the reviewer’s comments below, we perceive that the *specific concerns* announced above are mostly related to an incomplete discussion of the liquid cells that enable extremely high flow rates (described in Petruk *et al*, *Nanotechnology*, 2019 and repetitively referenced by reviewer #1; though effectively a single inlet device). We balanced the discussion of such systems emphasizing their benefits in the special application scenarios they were initially designed for, *i.e.*, ultrafast diffraction studies which require large field of view and benefit from tremendously high solution replenishment rates. At this point, however, it should be mentioned that, as Petruk *et al*. state themselves, "even a small flow rate of 0.1 $\mu\text{L}/\text{min}$ was found to be too high for STEM measurements", which invalidates most of the reviewer's subsequent arguments: if stopping the flow is required for imaging, the advantages of fast convective refresh rates vanish entirely as a radiochemical steady-state is established within milliseconds as commonly known (*e.g.*, Schneider *et al*, *JPC*, 2014), thus stopping the flow for imaging would be essentially equivalent to the static cell in terms of radiochemistry.

To keep up with the ongoing advances in microfluidic systems, we additionally referenced a *fresh-off-the-press* manuscript (Kunnas *et al*, *Ultramicroscopy*, 2023, Ref. 25 in revised manuscript) which recognises and validates the importance of diffusive transport for solution replacement and the necessity to have a short diffusion path from a connected reservoir. We additionally reference the latest review on LP-TEM methodology (Chen *et al*, *ACS Applied Nano Materials*, 2023; Ref. 28 in the revised manuscript), which gives a broader and more general overview of liquid cell designs.

As we will demonstrate below, we believe that the claims made with respect to the developed *diffusion cells* are valid, presented in adequate tone (compare comments of reviewer #3), and most importantly inspiring for a broad interdisciplinary research community.

To elaborate, the authors must address the importance of reducing window bulging in flow LPEM systems by connecting the outlet tube to an air-evacuated container. This adjustment will set the background pressure in the outlet of the nLC at approximately $P_{\text{outlet}} = 30$ mbar, assuming the use of pure water for simplicity. The assumption made in the simulations that $P_{\text{outlet}} = 0$ raises questions about its actual value in the conducted flow experiments.

There is a series of aspects in this concern. We are aware that these aspects reappear to some extent in the feedback from other reviewers, who will be referred to the following paragraphs where appropriate.

1) Importance of bulging in flow experiments in general.

The importance of bulging results from the fact that a thicker liquid layer deteriorates resolution. This is relevant when a large field of view is required, *e.g.*, for electrochemical experiments or for pump probe diffraction. At the same time, a large field of view assumes low to medium resolution, so, bulging is not restricting, yet of course limiting for this class of experiments. For high resolution studies, it is common to image in the corners of the windows, where the liquid layer thickness is close to nominal, and thus bulging is not relevant. Anyway, current developments in microchips design (see *e.g.*, Jensen, *Microscopy and Microanalysis*, 2014 (Ref. 23 in revised manuscript) or Koo *et al.*, *Adv. Mat.*, 2020 (Ref. 32 in revised manuscript)), provide solutions to obtain uniformly thin liquid layers across the field of view, as already pointed out in the initial manuscript. Thus, the struggle with bulging no longer seems to be an issue nowadays.

2) Relevance of bulging for the topic of this paper.

The authors do not quite see the relevance of bulging for the topic of the current work. We discuss mass transport mechanisms, which hardly depend on bulging and thus none of the claims of the paper may be significantly influenced when bulging is considered. Simulations confirming the limited importance of bulging were already presented in the Supporting Information of the initial manuscript (compare Fig. S10 on p. 15 in the revised manuscript). In the revised manuscript, the corresponding section was extended (in line with concerns of reviewer #3) both with experimental and numerical data demonstrating that replacement dynamics occur homogeneously in the entire viewing area (compare Fig. S6, S7 & S9 in revised manuscript) and do not change dramatically even at very conservative bulging estimates.

3) Misconception of the reviewer concerning outlet pressure (in simulations)

$P_{\text{outlet}} = 0$ used in the simulation is not an assumption and it has no influence on the outcome of the simulations. Since water is considered as an incompressible fluid and membranes are considered to be hard and solid, the background pressure does not have any influence on the flow, only the pressure difference between in- and outlet. A similar approach has been used in simulations for example in Petruk *et al.*, *Nanotechnology*, 2019.

4) Utilization of negative (relative to atmosphere) pressures in microfluidics system.

Negative pressures have been successfully tested and proven useful for reducing bulging in static and flow cells. To the knowledge of the authors, pressure-driven pumping systems are readily available and enable controlling in- & outlet pressure independently by pressurizing the inlet with gas and applying negative (relative to ambient) pressure (“vacuum”) to the outlet; however, pressure calibration of flow setups operating at negative (outlet) pressure is scarce and the experimental realization in flow setups – according to the author’s experience – is not that straightforward.

The benefit of using negative outlet pressures strongly depends on the overall working pressure regimes. With increasing gradients (several 100’s mbar, such as outlined by Petruk *et al.*, *Nanotechnology*, 2019), the dissolved gas used to pressurize the inlet solution becomes increasingly likely to nucleate and accumulate in gas bubbles along the flow path which can alter the hydrodynamic properties of the device. Geometric features such as sharp corners in the flow channel, which are typically present in LP-TEM flow systems, are known to facilitate the formation of gas bubbles. According to the experience of the authors (unpublished data), the flow starts showing instabilities, at outlet pressure as low as 400 mbar (-600 mbar relative to atmosphere) due to (partial) outlet line blockage by gas bubbles.

Another issue intrinsic to pressure-driven pumping systems is the altered composition of inflowing solution by pressure maintaining gas that may interfere with the processes studied. In said pumping systems, a gas is used to pressurize the inlet solution. In a recent publication, we have demonstrated

that saturating the flowing irradiated liquid with gaseous species strongly alters the radiolysis reaction network and therefore potentially causes beam-induced artifacts (Merkens *et al*, *Nano Express*, 2023; compare also concerns of referee #5). The chemical models that take this problem into account are not yet available (radiation chemistry of N₂ is not included in established radiolysis models).

The authors seem concerned about the $\Delta P = 300$ mbar calculated in [Petruk *et al.*, *Nanotechnology* 2019], a value that is substantially lower than the background pressure level of 1000 mbar – the typical value experienced by nLC in most experiments that were performed with flow LPEM systems with the outlet tube open to the lab atmosphere.

The authors emphasize that, until here, the reviewer’s argument is about pressure in the context of bulging. It should be clarified that the concern in the manuscript is mainly about the **relative** pressure difference between the in- and outlet (Δp) which is constant for fixed flow rate (compare Eq. 1) and has no relation to the pressure outside.

We would like to remind the reviewer that the 300 mbar reported by Petruk *et al.*, are for “nanochannels” of 1000 nm height. The pressure drop is expected to increase substantially when the height of the nanochannels reaches values comparable to those reported in the presented manuscript (< 200 nm) and acceptable for TEM/STEM imaging at decent resolution. Following the equation provided below by the reviewer, which shows 3rd power dependence on the channel height, the pressure developed in 200 nm height channel is expected to reach > 35 bar at the same flow velocity.

Nevertheless, the statement was clarified in the revised manuscript (compare p. 3 of the revised manuscript).

The manuscript appears to make an apples-to-oranges comparison by suggesting that diffusion can be more effective than convective mixing. Forced mixing should inherently be more efficient than relying solely on diffusion. The authors should clarify this potential misconception, particularly considering that the ‘true’ flow LPEM system presented by Petruk *et al.* offers sample refreshing rates about 1000 times higher than their proposed method.

We believe that there is a principal misconception of the reviewer(s) concerning the term “mixing” in microfluidic (in particular LP-TEM) systems. We will try to resolve this confusion and hope to thereby clarify that at no point in the manuscript such statement was made.

Considered independently, convection describes the directed/forced bulk motion of a fluid, while diffusion describes the random motion of a solute within it. The flow at the dimensions and velocities characteristic for LP-TEM liquid cells is in a deep laminar regime (Reynolds number is well below 1, while transition to turbulent takes place at about $Re \sim 2000$), *i.e.*, fluid elements move on parallel flow lines and no turbulences are expected to alter the composition of a travelling fluid element (= no convective mixing). In contrast, net diffusive transport occurs when spatial concentration gradients are present and lead to an equilibration of concentration with time (diffusive mixing). In other words, diffusion is the only mixing mechanism (there is no convective mixing) active in laminar flow reactors. This is why microfluidic mixers are generally such elaborated devices that aim to increase the flow path to allow the interdiffusion of mixing liquids at the microscale (see *e.g.*, <https://www.elveflow.com/microfluidic-reviews/microfluidic-flow-control/microfluidic-mixers-a-short-review/>).

In the manuscript, we are at no point stating that “diffusion can be more effective than convective mixing” in particular because we clearly realize above-described peculiarity of mixing in microfluidic devices. What we actually do state is: diffusive mass transport scales quadratically with lateral extension of the central nanochannel, while convective mass transport (not mixing) scales linearly; a reduction in dimensions eventually leads to a dominance of diffusive over convective mass transport at the same flow velocity; the geometry may be defined for which convective transport with all its drawbacks can

be eliminated, while diffusive transport will provide mixing rates not achievable in liquid cells governed by convection.

At this point, it is crucial to remember that the device described in Petruk *et al*, *Nanotechnology*, 2019, which to a large extent forms the basis of the concerns brought up by reviewer #1, is a single inlet setup, thus limited to convective solution renewal and incapable of performing mixing/replacement.

We have refined the wording where appropriate to better differentiate between the aspects of convective and diffusive mass transport.

Furthermore, the difference between refresh rate and mixing rate must be outlined - please see considerations about mixing in laminar flow. Besides high (mono)liquid flow velocity and thus high refresh rate, fast mixing/replacement is impossible to achieve in the single inlet design presented in Petruk *et al*, *Nanotechnology*, 2019, and the implementation of a design based on convective mixing would require sophisticated premixing channels and would therefore not necessarily enable fast mixing times. In the words of the reviewer, the introduced *diffusion cell* is an entire fruit salad, combining numerous benefits for solution renewal as well as mixing/replacement.

There is yet another consideration concerning Petruk's design, which prevents us from making a direct comparison with the ones in our manuscript. The fast flow system in Petruk *et al*, *Nanotechnology*, 2019 is designed for ultrafast pump probe experiments (including ultrafast electron diffraction) in a specially designed TEM setup. This type of experiment has particular requirements, which are different to general type LP-TEM experiments (*e.g.*, diffraction does not require fixation of the sample, while a large transparent area is desired) and thus comparison of the system developed by Petruk *et al*. to the general-purpose LP-TEM systems really would be an apple-to-orange comparison. It should be pointed out, as the authors of Petruk *et al*, *Nanotechnology*, 2019 have themselves arbitrarily pointed out, that their system does not allow (S)TEM imaging if liquid flow is activated ("The flow must be stopped [...] to achieve a decent resolution < 10 nm" (section 2.1 "Flow Simulations")), which automatically disqualifies it as a LP-TEM system for conventional TEM imaging in the majority of scenarios.

The pressure differential between inlet and outlet for liquid to transverse a rectangular channel can be calculated using the following expression:

$$\Delta P_{inlet-outlet} = 12\rho\nu ECL/CW (Spt)^3$$

where $\rho = 1 \text{ g cm}^{-3}$ is the density of water, $\nu = 0.892 \text{ mm}^2 \text{ s}^{-1}$ is the kinematic viscosity of water at 298K, and V is the volumetric flow rate. The remaining parameters define the channel geometry; CW is the channel width, Spt is the channel height, and ECL is the effective channel length. This expression can be rewritten as a function the refreshing time t as follows,

$$\Delta P_{inlet-outlet} = 12\rho\nu ECL^2/(Spt)^2 t$$

As the authors could see, increasing t from 1 ms (Petruk *et al*.) to 1 s would decrease $\Delta P_{inlet-outlet}$ a thousand times and make it rather negligible when ECL is in the order of 50 – 100 μm . Therefore, the authors should properly compare their design concept with those with 'true' flow systems. The expressions provided below are in good agreement with results from COMSOL Multiphysics.

We appreciate the demonstrated knowledge on the analytical description of laminar flow (which the reviewer could also find to some extent in the Supporting Information of our manuscript(s)). We completely agree with the reviewer in their description of the system by Petruk *et al*., however we do not quite see the relation of this analytical description to the topic of the current work.

We further do not quite see the subject of comparison. The work we have presented is focusing on the mixing dynamics of two solutions, not the replenishment rate. We have not found in the literature (available on initial submission) a single paper (besides our own article Merkens *et al*, *Ultramicroscopy*, 2023) neither describing nor calibrating mixing dynamics. Even taking into account the above

considerations on mixing in laminar flow, a "real" flow system such as that of Petruk *et al.* cannot have any physical advantage in terms of mixing. Moreover, assuming extremely low overall volumetric flow rates required due to the narrow nanochannel (compare 0.3 uL/min in Petruk *et al.*, *Nanotechnology*, 2019 to 50 uL/min = 3000 uL/h achievable in the presented work), direct flow cells with multiple inlets most probably would suffer from substantial delay time for liquid replacement as the volumes of supply and mixing channels are orders of magnitude larger than that of the nanochannel and thus the linear flow velocity there is extremely low. This does not apply to the closed cell design of Molhave and co-workers, where mixing is performed directly in nanochannels.

We understand that the reported system in Petruk *et al.* is a single inlet setup, thus only suited to perform (extremely) fast renewal in the NC. However, it has drawbacks which, to our understanding, can only be resolved by the proposed design of the *diffusion cell* as a whole: 1) high flow rate in the NC, probably flushes freely moving samples; 2) high flow rate in the NC, implies extremely low flow rates in the channels leading to the NC, thus, 3) in a mixing setup, *i.e.*, with a double inlet, it would be unlikely that mixing occurred inside the limited viewing area, as this is governed by diffusion, or 4) favour clogging if mixing is performed in large premixing channels due to low linear flow rates.

In this regard, the authors should also reference early groundbreaking work conducted by the teams of Bau [Grogan and Bau 2010 *J Microelectromech. Syst.* 19 885–94] and Miller [Mueller C et al. 2013 *J. Phys. Chem. Lett.* 4 2339–47], which, to the best of my knowledge, were developed independently of the offerings from the primary LPEM OEMs.

We are grateful to the reviewer for providing these valuable references, which are now included in the introduction (#27 and #26 in the revised manuscript). We have to underline that none of the systems described enable the mixing/replacement of liquids.

Lastly, the label “flow cell state of the art” in Figure 1a is misleading. This figure actually depicts a “conventional” nLC, consisting of two “flat” chips separated by a spacer. From an engineering perspective, these conventional nLCs are not configured to support true flow dynamics. Instead, as the authors themselves have pointed out, they function as diffusion cells. Their design, which features a large *wNC*, limits their efficiency in mass transfer and liquid flow mixing. These traditional nLCs prioritize cost-efficiency, aiming to minimize the number of nanofabrication steps required for production. In contrast, the authors propose a straightforward yet effective solution.

We recognize that “state of the art” is misleading and appreciate the reviewer’s suggestion to refer to the depicted setup as “conventional”. We have therefore changed this part of the document accordingly. We further appreciate the detailed evaluation of such conventional LCs by the reviewer, to which we mostly agree.

Their design adds only a single additional RIE step to the manufacturing process. However, it's important to note that the feasibility of incorporating this modification by existing LPEM OEMs hinges on whether it would infringe upon any existing intellectual property rights.

We agree to the reviewer and extended - within the scope of our possibilities - the methods section on wet-etching which was conducted in house. Regarding chips fabricated via RIE, they were obtained upon request from the manufacturer; we have no further information thereon that we could provide. A statement to reflect these limitations was inserted in the manuscript (p. 18 of the revised manuscript). According to our research, the presented *diffusion cell* concept is not in conflict with any intellectual property. If the reviewer knew any, we would be glad to be made aware of it. Nevertheless, intellectual property rights to our knowledge only prohibit commercial exploitation of an idea; the purpose of the paper, however, is to illustrate the concept and demonstrate its benefits for experiments.

Additional minor comments:

“(…) simulations were performed extending the tested range for the width of the nanochannel ($2 \text{ mm} < w_{\text{NC}} < 0.05 \text{ mm}$)”. If I understand correctly, w_{NC} was varied $2 \text{ mm} > w_{\text{NC}} > 0.05 \text{ mm}$.

The reviewer is correct, the typo was corrected.

Reference 31 appears to have an incorrect link.

The link appeared by mistake. The reference is corrected in the revised manuscript.

Please elaborate regarding the technological limit for 30 μm windows. How much does this limit depend on the material and thickness?

The authors want to point out that 20 μm wide windows were used to decrease the diffusion path to the minimum yet staying with commercially available chips. We do not produce the chips ourselves and are therefore not aware about technological limits. There is a limitation of the depth of the on-chip bypass, as the walls after RIE are vertical, while the groove forming the window from the back side is KOH-etched and has an inclination of 45° (schematics have been added to the Supporting Information, Fig. S1 on p. 9). Chips used in this work were fabricated in collaboration with the company Protochips and obtained as is. In-house chips (not reported in this paper) were fabricated using KOH etching, which ease the above restrictions and give a substantially smaller w_{NC} and a substantially deeper on-chip bypass.

Please clarify the difference in the flow path between the bathtub / onsite mixing setup shown in **Figure 2** versus the bathtub / onsite mixing setup shown in **Figure 4**. It appears that there are multiple inlets in the setup shown in **Figure 2** but only one inlet in the setup shown in **Figure 4**. The inactive inlet is discussed and illustrated in **Figure 5** but perhaps this can be clarified in figures which appear earlier in the text as well.

We appreciate the reviewer’s feedback regarding the confusion of flow paths in Fig. 2, 4 and 5. We included various changes to clarify flow paths throughout the manuscript. Most importantly, we included missing arrows indicating inactive inlets in Fig. 1 and 4, which were initially neglected for simplicity.

Reviewer #2 (Remarks to the Author):

We appreciate the reviewer’s contribution to the Nat. Commun. trainings initiative for Early Career Researchers by co-reviewing this manuscript.

Reviewer #3 (Remarks to the Author):

As a general remark to Reviewer #3 – he/she mostly criticises *Supporting Information*, which in the opinion of the authors is not a common practice, as *Supporting Information*, though being an inherent part of the manuscript, contains the information, which helps understand (the complexity of) the work in the main document, and may, in particular, contain experimental data, which were inconsistent or simulations which did not go entirely well.

Here in this article, the authors have proposed a novel design of TEM liquid cells that functions on the principle of diffusive mass transport, where they have fabricated the design based on their results from numerical simulations and have characterized the working regime.

The primary mixing time points and related discussions significantly depend on the relationships derived for the mixing time (and as mentioned on Page 7), and we notice here that the relationship derived is actually less absolute than the “mixing time”- claims outlined in the manuscript would suggest.

The authors are puzzled by these comments from the reviewer. We can only assume that “primary mixing time points” refers to the discussion of Fig. 3 and would like to point out that they are not calculated from the derived relationships. The reviewer should note that Eq. 1-6 are presented to describe the dependence of mixing time on different parameters and thus to discuss the diverse options to improve hydrodynamic properties. The first mixing times reported in the last paragraph on p.7 concerning Fig. 2 are actually simulated with validated finite elements models, thus they are absolute for the configurations they were obtained for – thus are the claims.

Additional sub-headers (“Model Development – Stationary Flow” & Model Development – Dynamics Solution Replacement”; p. 6 & 7, respectively) were included in the revised manuscript to separate paragraphs on model development from the illustrative quantitative description.

The reviewer is right in the sense that diffusion-controlled mixing times depend on diffusion length (w_{NC}) and coefficient (D); D itself depends on solute and solvent properties (see below); which for the most relevant LP-TEM scenarios (water and small solutes) will be in line with the claims. Note that $D = 6e-10$ m²/s used in the main draft (e.g., Fig. 3), is rather slow compared to many chemically relevant salts.

For example, in equation S12.13, the units aren’t satisfied. Also, the authors need to better describe the need to generalize the equation by eliminating terms such as viscosity, which deviates them from the authors’ heavy claims to be the fastest mixing TEM flow cell.

First, we would like to point out that equation S12.13 is denoting a proportionality; there is no need for units to be satisfied, since the proportionality constant (here: viscosity) carries units. Equation S12.13 is generalized to underline a linear dependence of t_c from the size of the channel. Minor changes to the text were introduced to clarify the purpose of Supporting Information section 2.4 (p. 5).

Concerning reviewer’s allegation on heavy absolute claims, authors would appreciate a reference describing a faster mixing cell for LP-TEM to confirm the relevance of the reviewer’s concern. We do not quite understand why considering viscosity to be constant in derivation of S12.13 may influence the claim of fastest mixing cell – it is fast by construction, not fast because of the special liquid which was used for validation. It should also be emphasized that for now water and water-based solutions are the **most relevant** liquids in practical LP-TEM research.

For these reasons, the authors are suggested to significantly tone down their claims (that sound very absolute), especially not to mention that they have developed a flow cell with the fastest/shortest mixing time when they have performed the simulations and experimentations only to a limited scope of sample regime (i.e., viscosity, the inclusion of particles, particle types, device dimensions, etc.). Under certain real-world conditions, the quoted mixing times would be much longer than the claims in the paper imply.

We outline below a comprehensive list of the mentioning of “fast mixing” in the manuscript:

- “Towards sub-second Solution Exchange Dynamics“ (title)
- “namely the possibility [...] for fast (within seconds) solution exchange dynamics” (p.4)
- “Even for the relatively low diffusion coefficient used in the simulations ($D = 6 \cdot 10^{-10}$ m²/s), τ can reach ~ 1 s for $w_{NC} = 100$ μ m” (p.10)
- “a virtual double-inlet diffusion cell hypothetically compatible with the Stream system []. Simulated solution replacement dynamics for such devices also pointed towards sub-second mixing times” (p. 15)
- “the fastest experimentally obtained mixing constants were ~ 2 s, which is ~ 2 orders of magnitude faster than previous results on default setups;²⁵” (p.17)

- “Simulations of optimized scenarios demonstrated the ability to achieve sub-second mixing/exchange dynamics when applying sophisticated experimental methodology.” (p.17)

In this list, we do not see exaggeration of the tone as mentioned by the reviewer. It should be once again pointed out, that water is the most important liquid in today's LP-TEM and thus time estimations made for water are the most relevant ones in this context – the scope of the **manuscript** is rather broad than limited. In any case, we have shown all the dependencies, including those from viscosity and diffusion coefficients and the numbers can be easily scaled if other (*exotic*) liquids and different solutes are used.

We have clearly defined the scope of the experimental conditions and simulations. The claims of the manuscript do not directly depend on viscosity for classical Newtonian liquids. The time constants will certainly be different when nanoparticles are considered due to substantially lower diffusion coefficients, but considering particles is beyond the scope of this paper; we will rather address the reviewer to a recent paper by Kunnas and co-workers [Kunnas *et al*, *Ultramicroscopy*, 2023; ref. 25 in revised manuscript]. The time constants would be even faster in many application scenarios, given that the contrast agent (PTA) has a relatively small diffusion coefficient (6e-10 m²/s) compared to many chemically relevant species. Device dimensions will influence the time constants, but this is exactly what the paper is about. It should be noted that there is not much play in dimensions in LP-TEM cells (see extensive section on “Device Fabrication & Design Rules”), they have to be in the order of a few millimetres.

We agree, that at certain conditions the mixing time will be substantially longer, while we do not quite understand, why at these complicated conditions (*e.g.*, higher viscosity) an alternative design should be better – the physics of flow and diffusion will stay the same in any design, we just propose the optimal one. As it was shown in Petruk *et al.*, *Nanotechnology*, 2019 (compare above), fast flow cells are not an alternative – as it might seem at first glance – as fast particle supply in the flow prevents their imaging.

In Figure 3 a, d, g, j, the x-axis is plotted in descending order while all other plots in the comparison are in ascending order. This change can be confusing to the reader.

The representation is chosen in-line with the manufacturing process, *i.e.*, decreasing w_{NC} , such that the *desired* trends, *i.e.*, decreasing Δp , v_c , Δt and τ , evolve from the left to the right. The positive way to think about this plot is that the size of the bypass, *i.e.*, $w-w_{NC}$, is increasing but plotting it like this would be even more confusing.

Given that the majority of reviewers did not report similar issues (#3 is the only reviewer who brought up the concern), we would like to stick to the current form of x -axes.

Further, to take into account of plots from Fig. 3, and its corresponding claims of mixing times, the authors are suggested to add more data points (especially at short mixing times where there are too few) to observe a smooth transition/curve through the data.

The plots illustrate the dependence of the depicted parameter in the whole range of accessible experimental and structural parameters for the virtual prototyping procedure; therefore, we don't see the need for adding more datapoints. The purpose of these plots and the simulations behind is to screen the parameter space over a broad range (1 order of magnitude) and to validate dependences brought up in the beginning (Eq. 1-6; 2 orders of improvement with respect to mixing time constants).

Furthermore, it should be noted that the underlying models diverge from the physical prototypes that can be and have been fabricated for several reasons. The benefits of spending resources thereon are limited if not missing.

It also remains unclear to the authors where the reviewer locates the regime of *short mixing times*, and how *many* datapoints should be added to satisfy the request. In the case the reviewer refers to *sub-second* time constants, these were only discussed in context of the hypothetical direct flow/onsite mixing configuration “hypothetically compatible with the Stream system”. It is therefore different to the configurations depicted in Fig. 3, which were derived from Poseidon Select and Poseidon 200.

In Line 344, the authors have mentioned the inclined groove. The authors are suggested to provide detailed dimensions of the groove since it is important for the TEM measurement considerations.

We would like to point out that the inclined groove has nothing to do with TEM measurements. It is a groove at the back side of the commercially available chips which provides access to the window. Inclination comes from the technological process, which uses KOH etching resulting in 45° walls. The authors have nothing to do with this and up to our knowledge most if not all designs of the LP-TEM cells have this characteristic.

In the experimental testing of diffusion cells, the authors have briefly mentioned the bulging of the nanochannels. Since the mixing time period and its claims therein in the entire publication depend on these dimensions, and as mixing by diffusion is directly proportional to the channel dimensions, it appears necessary that the authors perform further experimental and numerical analyses on the same.

We would appreciate to learn, what the reviewer exactly meant by diffusion being proportional to channel dimensions? Diffusive flux does not depend on the channel dimensions, but only on the gradient of the concentration, thus the reviewer's concern is to a large extent invalid. Diffusion time in turn is a square function of the diffusion length, see eq.6 in the main manuscript (but not the height, which is the only measure which changes upon bulging), and the main message of the paper is that nowadays it is technologically feasible to manufacture the structures allowing for diffusion time of the common solutes in water reaching values below 1 s.

In turn, we thank the reviewer for raising the intriguing topic of window bulging. In the revision process we have dedicated major resources to present an in-depth study on the (ir)relevance of bulging in diffusion cells. The main additional observations comprise:

1) Effect of bulging on mixing time constants is relatively small compared to other effects.

Fig. S10 in the revised manuscript (p. 10 in *Supporting Information*) evaluates the effect of bulging in context of different flow rates, flow channel geometries and inlet boundary conditions. Despite a notable increase of both time constants due to bulging, their variation is lower with respect to other parameters investigated, in particular the gradual inflow gradients of contrast agent solution.

2) Mixing rate is uniform across the viewing window.

Fig. S6, S7 and S9 in the revised manuscript (p. 12-14 in *Supporting Information*) depict the variation of concentration curves (simulated & experimental) and extracted time constants at different positions across the imaging area. The replacement time constants are independent on the position across the window within the error bars (Fig. S7; p. 13 in *Supporting Information*).

3) Different Effects of volume increase are counteracting.

To avoid potential misinterpretation of the complex 3D models based on realistic flow channel geometry, simplified (2D axisymmetric) diffusion models were implemented to obtain a clear view on the effect of bulging on diffusive flux. The model is introduced in section 4.2.3. of the *Supporting Information* (p. 16) in the revised manuscript. Simulations show that due to the increased liquid volume in the imaging area because of bulging, a high concentration gradient is developed, which leads to substantially increased diffusive flux. This increase however does not completely compensate for increased volume to be filled and the concentration equilibration is delayed and more gradual than with flat membranes (refer to Fig. S12 & S13 for details; p. 17).

With respect to numerical analysis, for bulging, it would be expected to perform time-resolved studies that include fluid-structure interaction to accurately model the channel shape during bulging, and not just truncated shapes (that currently seem arbitrary) as described by the authors in Fig. S5.

The implementation of even more sophisticated models, *e.g.*, fluid-structure-interaction, to reflect experiments more accurately, is indeed a fascinating challenge, which the reviewer has addressed. However, assuming the comment above, the authors do not see the necessity to perform a detailed study of the shape of the bulged windows. The *simple* but conclusive model(s) presented in the Supporting Information (p. 14ff) shows only a weak dependence of the mixing times on bulging. We should also point out that the bulging value of 2 μm used in simulations (provided by the manufacturer; Ref. 35 in the revised manuscript) is very conservative, so that direct measurements of bulging for 20 μm windows give values well below 2 μm . Therefore, we believe that the suggested work would exceed the scope of the manuscript. We redirect the reviewer to some papers (*e.g.*, Petruk *et al*, *Nanotechnology*, 2019 or Keskin *et al*, *Nano Letters*, 2019) where the shape of the bulged membrane was calculated and measured, but not yet considered for the flow simulations.

We further would like to point out that the model refinement in context of Fig. 4 and 5 was not conducted to precisely reproduce experiments, but rather to understand possible aspects explaining deviations from the initial models. As we have demonstrated both in the Supporting Information and in the previous manuscript (Merkens *et al*, *Ultramicroscopy*, 2023), mass transport dynamics are affected by multiple, often interconnected, aspects (in Poseidon 200 setup, this is most importantly the poor reliability of the liquid cell assembly). The expected outcome of proposed extended simulations and experiments would hence remain questionable.

For experimental analysis, bulging of channels can be characterized at various flow conditions, such as flow rate (& fluid conditions such as viscosity, particle density, etc.) using tomographic or detailed spectroscopic analysis. Maybe there are also options to accurately map out the bulging deformation at different relevant conditions using the TEM signal.

The reviewer is certainly right that such options exist. However, we would like to underline here, that bulging is out of the scope of the present work, and since the irrelevance of bulging for the claims of the paper was confirmed, it is not clear why we should study this phenomenon further in this manuscript.

In the hydrodynamic characterization using spectroscopy, the authors have performed the analysis at 10 Hz, while discussing about such short timestamps, the authors are encouraged to collect data at a higher frequency.

The experimentally measured time constants are of the order of seconds (meaning saturation is achieved in tens of seconds). Acquisition is performed at 0.1 s intervals. Having ~ 100 points for curve fitting is more than sufficient from any point of view. We therefore do not see the need to repeat the experiments as the reviewer suggests.

In Fig. S7b, the authors have provided limited results, and more data points would be required.

The provided data was obtained for four KMnO_4 solutions with different concentrations and depicts the corresponding absorbance at the maximum wavelength over a range from 0.2 to 2, *i.e.*, covering the full accessible range (nearly approaching the detection limit indicated by the *grey area* indicated in Fig. S14a of the revised manuscript, *Supporting Information* p. 19). As can be seen from linear regression in Fig. S14b, the fitting of the data points is very good, and the results *complete*.

Overall, The authors have only briefly touched upon the bulging using spectroscopic analysis, further analysis needs to be performed on both experimental and numerical analyses.

We must disagree with this comment. Due to the limited relevance of bulging for replacement dynamics (compare previous responses), which is discussed in depth in context of numeric simulations and experiments in the revised **manuscript**, we find satisfying the spectroscopic analysis presented and hope that the reviewer will share this opinion.

Additionally, the bulging should be significantly less in the spectroscopic setup, as the device stays at atmospheric pressure, and thus the acting pressure difference responsible for bulging is the one determined by the flow resistance of the cell (*i.e.*, a few mbar), but not ~ 1 bar as in real TEM experiment.

In Fig. S9, d, and e, it is clearly observed that there is a significant deviation between experimental and numerical analysis, thus in line with the previous comments.

We would like to thank the reviewer for pointing out this discrepancy. The plot for Δt appeared by mistake and we removed it in the revised manuscript to avoid confusion. The system described in previous Fig. S9 (S16 in revised manuscript) is different from the commercial TEM holder in the part of supply lines (though the microfluidics part formed by & around the MEMS-chips is identical). The delay time is mostly determined by solution travel time in the supply lines and thus may differ from the one obtained from simulations, where a TEM holder geometry has been implemented. In turn, τ is defined by hydrodynamic properties of the microfluidics part defined by the MEMS-chips, mainly the NC, and thus is more indicative. Simulated values for τ in Fig. S16d (previous S9e) are indeed very close to experimental observations, which indicates significantly lower bulging as compared to TEM experiments.

Hence, the authors are encouraged to further investigate their numerical analysis and reevaluate their experimental results in Fig. S9, S10, and S11 in the supplementary and in the main draft Fig. 4 d, e, f, g, h, i.

We thank the reviewer for this comment. However, we want to outline that a great deal of work has been put into refining numerical simulations as clearly illustrated by Fig. S10 (already included in the initial manuscript), where the *trivial* model, *i.e.*, non-bulged NC, the bulged model, the trivial with gradual inflow gradients (compare effect of diffusive broadening in Fig. S5 of the revised manuscript) and the bulged model with gradual inflow gradients are analysed. The figure clearly reveals that a combination of both can semi-quantitatively explain the deviations between simulations and experiment: the increased volume due to bulging has negligible effect on the shape of the resulting time constants (“bulged” is shifted above “trivial” by same value independent of flow rate in Fig. S10 of the revised manuscript), while a gradual concentration increase at the inlet propagates stronger with decreasing flow rates. Further effects of, *e.g.*, insufficiently precise reproduction of the *realistic* channel geometry and/or unprecise liquid cell assembly may cause deviations; a lengthy discussion thereon was already presented in Merkens *et al.*, *Ultramicroscopy*, 2023, as referenced in the presented manuscript.

In the fabrication of ex-situ Sample holders (6.1), the authors have mentioned about the Poseidon Select sample holder as the primary-based holder for further fabrication and assembly, and thereon for numerical, analytical, and experimental analyses. Hence, it is highly recommended to provide the patent number of the device and its dimensions of it. If not provided, it would go against the requirements of replication of the results by other users based on this paper. Also, it would be necessary to provide the direct flow/mixing onsite mixing setup.

The virtual model of the Poseidon Select sample holder was obtained from the company Protochips in the scope of a previous project (Merkens *et al.*, *Ultramicroscopy*, 2023) under a non-disclosure agreement (NDA; as reported in section 6.1 in *Supporting Information*). The authors have no further information about the patents held by the manufacturer. With the base model being protected through NDA, we unfortunately cannot share more details on it, nor the setups derived from it unless upon reasonable request.

We have also claimed, and this claim is supported by solid simulation sets, that the conclusions of this paper do not depend on the design of the holder itself but can be reproduced on any available holder independent of the brand just by using chips with an on-chip by-pass channel. All the data for reproduction of such design is provided.

In nanochannel mixing studies, it is quite common to investigate the clogging parameters by studying using various particle densities, shapes, and sizes, but no such experimentation, numerical analysis, and discussion have been performed in this article.

We would appreciate a single literature reference for such study made on LP-TEM systems. The reviewer raises concerns that are to a large extent irrelevant at the current state of LP-TEM.

We would like to make the reviewer aware of the practical implications of his/her comment. To our understanding, clogging parameters are typically investigated for disposable/low-cost nano-/microfluidic mixing devices. Notably, the costs of LP-TEM flow reactors can easily reach a hundred thousand dollars, which is why most EM groups possess 1 device only. Per definition, clogging tests result in the clogging of (part of) the flow device, which requires unclogging for multi-use devices. Unclogging LP-TEM sample holder is a highly challenging task that often requires dismantling the entire device, which is solely done by the manufacturer. As we have learned from previous *unintentional* “clogging tests” this procedure is both cost- & time-intensive; we do not intend to repeat that procedure deliberately.

Further, we do not think that clogging is an issue in conventional LP-TEM flow experiments given that most of the research is conducted in the regime of low solute concentrations. We therefore do not expect clogging to be a major issue that should be discussed in the context of this work.

Accordingly, we believe that the immense amount of work (and cost) implied by the systematic studies requested by the reviewer are disproportionate and strongly exceed the scope of this manuscript.

In the methods section, in Line 548, the authors have mentioned that both chemical and reactive ion etching strategies are utilized, while the details of reactive ion etching aren't described, nor the patent number has been provided (in case of IP protection). The authors are encouraged to provide a detailed fabrication procedure of the chip since the paper revolves around the development of a new liquid cell for TEM. Without more details, the value for the readers would be severely impacted due to the difficulty of and time needed for replicating the devices or results.

We agree to the reviewer and extended - within the scope of our possibilities - the methods section on wet-etching which was conducted in house. Regarding chips fabricated via RIE, they were obtained upon request from the manufacturer; we have no further information thereon that we could provide. A statement to reflect these limitations was inserted in the manuscript.

In the convection-diffusion model, the authors have assumed zero pressure at the outlet, which raises a major query and integrity of the entire numerical analysis performed in this manuscript, as obtained zero pressure is nearly impossible in such a TEM flow setup, and considering zero pressure creates higher momentum thus resulting in shorter mixing time and reduced bulging, which is quite not the case.

We appreciate the reviewer's comment regarding (outlet) pressure in flow simulations. In fact, the applied strategy is a common practice in computation fluid dynamics modelling. In forced flow, the relative working pressure gradient (between in- and outlet) is the only relevant parameter (*i.e.*, absolute pressure is irrelevant), if the fluid is considered to be incompressible and the channel walls are considered to be rigid. For more details, we refer the reviewer to the extensive replies made to reviewer #1.

Together with the feedback of the other reviewers, we believe that confusion arises due to the term “operating pressure” which was used to denote the pressure difference between in- and outlet in experiments on several occasions throughout the manuscript. Where appropriate, the wording was improved to avoid confusion in the revised manuscript.

Also, the authors are encouraged to provide why data points have been collected experimentally every 0.2 s (Line 584) without any scientific conclusion, which is very close to the authors' claims, thus raising concern about the experimental data integrity.

The fastest experimental time constants were 2 s, which means that the entire dynamics of the solution replacement takes place in a time span of a few to a few tens of seconds. For a time span of 10 s, and acquisition frequency of 5 Hz, which is what can be reasonably achieved with the microscope hardware, this results in 50 datapoints per curve. This is more than sufficient to accurately fit experimental curves, as can be seen from the large number of examples in Fig. 4d & e, which clearly justify its selection. We are surprised that the reviewer questions the integrity of the entire experimental data on this basis.

On a minor note, there is a small typo in Line 606.

We thank the reviewer for finding the typo, and we have corrected it.

The authors are encouraged to provide details on how the sample was prepared, how the diffusion coefficients were selected, and their references.

We thank the reviewer for noting the missing reference for the diffusion coefficient of the optical dye. We apologize for the flaw and have included the corresponding reference in the revised manuscript. We further added the reference for PTA (#25 in initial manuscript) on all occasions where the value is mentioned to prevent further confusion.

As for the sample preparation and selection criteria of contrast agent, the reviewer is referred to follow the indicated reference to the previous manuscripts (Ref. 29 in the revised manuscript), which establishes the contrast variation method in detail.

“The diffusion coefficient [...] $D = 1.3 \cdot 10^{-9} \text{ m}^2/\text{s}$ [in numeric studies was selected] to reflect more realistic scenarios of diffusing small molecules.”, already stated in the manuscript; the missing reference was added.

In this article, the authors have obtained at a single point in the nanochannel to get the mixing ratio, which doesn't include the distribution within the entire window. Thus, it would be necessary to provide a detailed analysis through the entire window spatially and temporally, where the authors would be encouraged to provide an analysis of the flow range, particle distribution, particle density, viscosity, and mixing ratios.

We are glad to inform that such analysis had been performed. No significant variations were found across the accessible area of the window, which is why the data were not included in the original manuscript. We included the requested data (both simulation & experiment) in the revised manuscript (Fig. S6, S7, S9). Given that no significant deviations were observed, we do not see further benefit in testing additional parameters (as suggested by the reviewer).

In context of the reported data set, there is one crucial realization to be made. The dynamics of the solution replacement were experimentally tested in the lateral center of the imaging area for symmetry considerations. However, directly in the centre the bulging is most pronounced, and therefore typically discarded for LP-(S)TEM imaging. With insignificant variation of time constants across the viewing area detected, this methodological discrepancy does not affect the generality of the obtained time constants & claims.

Also, in order for this device to be considered in the future by other users, it would be expected to provide universal relationships and a database to determine the mixing time points and corresponding fluid and flow conditions.

We invite the reviewer to compare his/her request against the available information on the properties mentioned for comparable setups. In fact, to our knowledge, they are inexistent! Not even commercial providers – for which one might expect a product to be fully calibrated – provide the requested characteristics. In contrast, the aim of the presented manuscript is not to precisely characterize and commercialize a product, but to introduce a new concept that drastically improves hydrodynamic properties paving the way for exciting new experimental capabilities, heavily awaited by the LP-TEM

community as it “*solves some of the most important issues in LP-TEM*“ as apparent from the feedback of reviewer #5, to some extent also #1 and #6, but still leaves room for further improvement.

In conclusion, this manuscript demonstrates an interesting development in the liquid cell TEM field, but it needs major revision with respect to experimental results, numerical analysis, analytical derivation, and explanations prior to further considerations.

We are pleased that the reviewer finds the work interesting and believe that we have addressed all concerns, so the reviewer approves publication of the revised manuscript in Nat. Commun.

Reviewer #4 (Remarks to the Author):

We appreciate the reviewer’s contribution to the Nat. Comm. trainings initiative for Early Career Researchers by co-reviewing this manuscript.

Reviewer #5 (Remarks to the Author):

The authors report on a novel liquid flow cell architecture for two liquid flow holder systems from Protochips utilized in liquid-phase TEM. Therefore, they developed a suitable layout by flow simulations, which is in a second step fabricated and tested in operational conditions. Their concept strongly reduces the flow resistance by bringing the channel inlet closer to the viewing area. This is achieved by modification of liquid cell chips by reactive ion etching. The novel concept is shown to strongly reduce solution exchange times while maintaining a purely diffusive solution exchange within the viewing area. In my opinion, the idea is brilliant and should be published after a minor revision as it solves some of the most important issues in LP-TEM that are so far strongly limiting its performance, repeatability, and versatility.

We are delighted about the positive perception of the idea by the reviewer and share the vision of relevance for the field of LP-TEM. In fact, we believe that the strongest benefit of the novel design is the comprehensive improvement of a broad range of hydrodynamic parameters with promising implications for a series of different LP-TEM flow experiments relying on efficient solution renewal, replacement & mixing. Despite being experimentally verified on a Protochips system, the concept is general and is directly applicable to other (multi-inlet) holders, either commercial or customized.

- In the modified LC, bare silicon (plus a native oxide layer) is exposed to the sample solution. This should have an influence on the stability and, thus, substances compatible to the chip. I don’t expect this to be a problem as the structuring using photolithography and reactive ion etching can easily be integrated into an optimized fabrication process in which silicon is structured before silicon nitride deposition. Nevertheless, for the prototype version, a comment should be added into the SI section.

We thank the reviewer for pointing out this important issue. It is true that the simplest possible fabrication route was selected to speed up the prototyping, but we should have commented about the modified chemistry of cell. This is corrected in the revised manuscript (p. 9 of *Supporting Information*).

Furthermore, in the methods section I think there is a RIE step for SiN removal missing between photolithography (this is only the structuring of the photoresist and not of the subjacent layer) and bulk micromachining.

The reviewer is totally right in terms of classical photolithography. Even though technically the laser is doing photolithography, it differs from the terminology established in the scientific community. To avoid confusion, we replaced the term by “laser sublimation” (p. 18).

- The authors mention that removal of radiolytic products is strongly enhanced due to the high exchange rates. As this is a very important aspect from an experimental point of view, there should be some more discussion.

We are delighted to hear about the profound interest of the reviewer in the removal of radiolytic products in *diffusion cells*, as we share the view of radiolysis being a highly relevant topic in LP-TEM research. In fact, dynamics in experiments are triggered by 1) external stimuli, *i.e.*, mixing dynamics, and/or 2) beam-related effects. Being strongly intertwined in most state-of-the-art experiments, we are convinced that it is crucial to separate both aspects in fundamental studies as the presented manuscript is. Since this manuscript is primarily dedicated to optimized mixing dynamics addressing a rather broad audience, we believe that extending the discussion on radiolysis will exceed the scope of the manuscript, considering that research on radiolysis is extremely vivid and complex and often strongly depends on the radio-chemical peculiarities of a given sample/irradiation scenario.

Based on the very targeted curiosities expressed by the reviewer below, we are more than happy to address the interests expressed based on preliminary unpublished results that we have obtained by implementing conclusions of this work in our previously established 1D model for radiolysis (Merkens *et al*, *Nano Express*, 2023; derived from Schneider *et al*, *JPCC*, 2014). The reviewer’s curiosities are addressed point-by-point below.

How does the novel architecture influence gas bubble formation (e.g. H₂), *i.e.* to what extend is a bubble formation delayed in comparison to conventional setups. I assume this can be estimated by comparing the generation of H₂ at a given dose rate with the removal rate by diffusion at a given flow rate up to a critical supersaturation value at which a bubble formation is expected.

Based on the implemented 1D reaction diffusion model, we can confirm a beneficial impact of the *diffusion cell* for the removal of gaseous species due to a series of effects. The first benefit results from opening the boundary: while in a closed cell geometry the self-scavenging of the radiolytic reaction network is incomplete, *i.e.*, some species such as H₂ increase in concentration with decreasing w_{NC} ; all (gaseous) species decrease in open cell configurations, as they can diffuse out of the central nanochannel. The second benefit results from decreasing w_{NC} . When decreasing w_{NC} from 1000 to 100 μm , concentrations of H₂ and O₂ decrease by approximately 10% but are expected to decrease by up to 50%, respectively 70%, for narrower NCs (< 10 μm).

For all tested conditions, concentrations were below critical supersaturation concentration (0.7 and 1.22 mM, respectively).

Another aspect to gas bubbles is their removal, in particular in electrochemical (EC) experiments. As we are demonstrating in a related manuscript (Ref #37 in revised manuscript; preprint available here: <https://doi.org/10.21203/rs.3.rs-3660145/v1>) removal of gaseous species is strongly facilitated by the novel *diffusion cell* design as proven in various experiments.

- How does the new architecture influence the redox chemistry (*i.e.* solvated electrons, H and OH and HO₂ radicals)? Is it comparable to conventional setups, or does it facilitate removal of reactive species?

In contrast to the gaseous species from above, redox-relevant species are mostly highly reactive species, *i.e.*, they tend to react directly within the irradiated area and are rather difficult to be removed through any means of mass transport.

Our preliminary simulations indicate that the concentration variations of the mentioned species remain within < 10% when w_{NC} is reduced from 1000 to 100 μm , although they eventually increase (*e.g.*, the

solvated electron & H radical) due to the altered self-scavenging (compare reduced [O₂] previously identified as main scavenger for the H radical; *e.g.* Merkens *et al*, *Nano Express*, 2023).

A key point to note is that as w_{NC} decreases, there is a clear and steady change in the concentration of the redox species, unlike in closed cell setups. In latter setups, trends are more unpredictable, similar to what was observed in flow-controlled scenarios with varying flow velocities (Merkens *et al.*, *Nano Express*, 2023). These findings indicate superior control over the reaction network through liquid cell design.

- In the same regard, to what extent is the acid-base chemistry (i.e. H⁺ and OH⁻) influenced?

We appreciate the interest of the reviewer in acid-base chemistry and accessed our 1D reaction diffusion model in this regard. The reviewer may be familiar with recent works of Fritsch *et al* (Fritsch *et al*, *J. Phys. Chem. Lett.*, 2023) accessing acid-base chemistry in terms of a radiation-adjusted parameters (π^*) given that external production breaks the pH equilibrium. With available simulations (initial pH = 7) in the range of relevant $w_{\text{NC}} = 100 \mu\text{m}$, we found π^* changing sign from -0.001 to +0.4 indicating a change from slightly acidic to slightly basic environment when increasing [O₂] to 1.22 mM. More relevant for flow-based scavenging methods is the comparison to closed cell setups for which significant differences become increasingly relevant with decreasing $w_{\text{NC}} \leq 100 \mu\text{m}$.

We acknowledge that the conditions tested may not be the most appropriate as the radiolysis network remains close to neutral under the moderate irradiation conditions applied ($\pi^* = 0$). Evaluating more demanding solution compositions and more extreme irradiation conditions is the starting point for a follow-up project currently being developed.

Continuing the previous concern, but much more in line with the actual scope of the manuscript, is the ability to quench beam-induced effects. We performed time-dependent simulations, in which the [O₂] at the outlet was sharply ramped from 0 to 1.22 mM (= saturation concentration). For $w_{\text{NC}} = 100 \mu\text{m}$ and 1 μm beam radius, all reactive radiolytic species change concentration (by nearly 0.5 orders of magnitude) in response to the increased [O₂] within only a few seconds after ramping the boundary conditions. As can be seen from the discussion in the manuscript, control over quenching can be improved by further reducing the NC extent.

- The authors state that the reduced flow resistance positively affects window bulging as less pressure is expected to reduce the outward bulging of the LC membranes. As window bulging is neglected in their simulations,

As clarified in the replies to previous reviewers (refer to *e.g.*, p. 9 of this document), bulging was approximated in those simulations aiming at reproducing experimental values.

two questions remain: Firstly, how does the complete chip bulge after assembly? In the shown configuration, spacers of 150 nm are in the four edges of the chip whereas the nanochannel has no spacer. Depending on the metal lid holding the chips in place, I would expect an outward bulging of the whole chips due to the remaining positive pressure separating both chips and probably determining the true diffusion channel height to be larger than the spacer thickness.

These considerations are correct, but they are equally valid for unmodified chips. The depth of the on-chip bypass is relatively small as compared to chip thickness (10 μm vs 250 μm), so we do not expect notable decrease in chip robustness and thus increased bending in comparison to unmodified chips.

Secondly, is the window bulging simulated in SI section 5.2. comparable to experimental values?

As already discussed in detail in response to reviewer #3, we think that the volume of the bulged area is a sufficiently good/conservative approximation of experimental bulging reported by the manufacturer (Ref 35 in revised manuscript). The considerations behind bulging estimations are discussed in more detail in the revised manuscript. The experimental setups required to determine bulging and related

pressure differences were outlined more accurately in a previous manuscript (Merkens *et al*, *Ultramicroscopy*, 2023; Ref. 29 in revised manuscript).

Reviewer #6 (Remarks to the Author):

The work by Merkens et al. presents an advanced silicon nitride liquid cell design that improves mixing during liquid phase transmission electron microscopy experiments. While the results are important and noteworthy for the LP-TEM community, they are more appropriate for a journal that specializes on microscopy. The work would become more significant in the field, and appropriate for Nature Communications, by utilizing the improved cells to study a problem that was not practical to study before with LP-TEM.

We partially disagree with this comment. In fact, the novel *diffusion cell* design does not solely improve mixing properties of the LPTEM flow system, but a broad list of hydrodynamic properties, as we emphasized in the manuscript. We believe that the versatility of the *diffusion cell* concept prohibits an adequate illustration of any foreseen significance across all diverse application areas. Therefore, we have conceptualized this manuscript as a comprehensive exploration of the improvements (if it was about mixing only, the last two columns in Fig. 3 would have been sufficient) which – if presented to a broad interdisciplinary audience such as the readership of Nat. Commun. – will stimulate high-quality research that we can neither conceive nor realize within the scope of a single research article alone.

Nevertheless, we are glad to report novel experimental work unambiguously demonstrating the significance of the presented manuscript as well as the versatility of the novel *diffusion cell* concept. As indicated in the letter to the editor and already announced in the original manuscript, a separate manuscript was submitted to Nat. Commun. showcasing the benefits of the *diffusion cell* concept for solution renewal during electrochemistry experiments in detail (Ref #37 in revised manuscript; preprint: <https://doi.org/10.21203/rs.3.rs-3660145/v1>). In addition, we present in the Supporting Information of the revised manuscript (section 7, p. 24ff) a solution replacement experiment capable of replicating timeframes of *ex situ* processes of colloid aggregation/disaggregation. To the best of our knowledge, the displayed experiment represents the first example ever of process/mixing times in the regime of seconds observed in LP-TEM on a sample that was deposited prior to the assembly of the LP-TEM flow reactor. The artifacts still observed in the image sequences, which are associated with radiolysis, indicate the limitations of the rapidly performed illustrative studies and call for further development of quantitative methods in LP-TEM, to which this work contributes, if not makes it possible.

The conclusions of improved flow are supported by the data analysis, and there are no substantial flaws in analysis, interpretation, or conclusions.

We appreciate the positive evaluation by the reviewer.

The methodology is sound, but there is minimal experimental data in the manuscript compared to other manuscripts in Nature Communications.

We would like to highlight that the introduced *diffusion cell* design is the outcome of half a decade of development. Fortunately – because this is the motivation behind our work – the experimental standards in LP-TEM have substantially evolved during this time. While initially impetuous qualitative observations were hardly questioned, quantitative methods are increasingly being integrated into the methodology with the aim of achieving correlative results. We assure the reviewer that we have collected numerous ideas for experiments during this time, which have now finally become possible and will be explored in follow-up projects using established standards.

In addition, we would like to emphasize that the manuscript is primarily not about experimental results, but about a novel concept that will advance the field and stimulate new studies that were not previously possible.

There is not enough detail to reproduce the manuscript as indicated by the lack of RIE information in the fabrication procedure.

The authors are grateful for this comment. We want to emphasize that two fabrication procedures were brought forward to modify chips relying on RIE and wet-etching, respectively. The RIE approach relies on well-established technology held by industrial collaborators at Protochips Inc., for which we have no further information, whereas the wet etching approach was developed in-house. The wet-etching process is described in detail in the *Methods* section allowing the fabrication of *diffusion cells* by interested readers.

While RIE leaves vertical edges in the etched chips as standard, wet etching results in bevelled edges, which has a positive effect on the hydrodynamic properties as it reduces sharp edges in the flow channel. The hydrodynamic characterization reported in the main manuscript to illustrate the *diffusion cell* concept was performed exclusively with RIE-fabricated chips due to superior control over the fabrication process, but can in principle also be obtained with wet-etched chips.

Throughout the revised manuscript, explanations have been included to clarify the status of the various manufacturing processes.

Find attached below, the list of changes to the main manuscript and the supporting information file.

Main Manuscript

Page #	Type of revision	Changes made
1	Authors List extended	 • Including Joscha Kruse,^{1,5} Maiara Aime Iriarte-Alonso,^{1,6} • Adding TECNIPESA IDENTIFICACION SL, Parque Empresarial Zuatzu, Edificio Donosti 1-3, 20018 Donostia-San Sebastián, Spain
2	Wording Improved	“The solution replacement dynamics achieved within seconds already matches the mixing timescales of many ex situ scenarios, although further improvements are possible.”
3	Wording improved; logics refined	“In LP-TEM flow reactors, [...] assembly of nanoparticles through solution mixing/replacement. [...] This versatility, combined with the ability to incorporate the stimuli mentioned above, ⁵ distinguishes flow systems from recently proposed static mixing cells. ¹⁴ ”
3	Discussion balanced	“Such flow setups with extremely fast sample renewal rates have proven to be particularly promising for special applications, e.g., pump probe imaging & diffraction; however, they come at the cost of increased operating pressure gradients (~300 mbar were reported for 1 μm high channels) ²⁴ , eventually resulting in increased bulging & the risk of window rupture, flushing of the sample, and limited solution mixing/replacement capabilities.”

3	Recent Reference added	 Ref #28: Chen, Y., Yin, K., Xu, T., Guo, H. & Sun, L. Characterization of Nanomaterials Using In Situ Liquid-Cell Transmission Electron Microscopy: A Review. ACS Appl Nano Mater (2023).
3	References updated	 Previous Ref #12 (Nielsen, M. H. & Yoreo, J. J. De. Liquid Cell TEM for Studying Environmental and Biological Mineral Systems. Liquid Cell Electron Microscopy 316–333 (2017)) was removed, and replaced with correct reference (Ref #11 (Nielsen, M. H., Aloni, S. & De Yoreo, J. J. In situ TEM imaging of CaCO₃ nucleation reveals coexistence of direct and indirect pathways. Science (1979) 345, 1158–1162 (2014).) in updated manuscript; previously Ref #29)
4	Wording improved	 “solution replacement [...] solution replacement” “summarized”
4	Recent reference added	“Just recently, the importance of diffusion for solution exchange, in particular replacement, was recognised by Kunnas and co-workers outlining the necessity for short diffusion paths between reservoirs of fresh solution to the IA.²⁵” (Ref. #25: Kunnas, P., de Jonge, N. & Patterson, J. P. The effect of nanochannel length on in situ loading times of diffusion-propelled nanoparticles in liquid cell electron microscopy. Ultramicroscopy 255, 113865 (2024).)
5, 14	Figure updated	 Arrows indicating flow path added
5	Reference added	 Ref #27: Grogan, J. M. & Bau, H. H. The nanoaquarium: A platform for in situ transmission electron microscopy in liquid media. Journal of Microelectromechanical Systems 19, 885–894 (2010). Ref #26: Mueller, C., Harb, M., Dwyer, J. R. & Miller, R. J. D. Nanofluidic cells with controlled path length and liquid flow for rapid, high-resolution in situ electron microscopy. J Phys Chem Lett 4, 2339–2347 (2013).
6, 7	Subheader added, readability improved	 “Model Development – Stationary Flow” “Model Development – Dynamic Solution Replacement”
10	Typo corrected	“2 mm > w_{NC} > 0.05 mm”
12	Wording improved	“was chosen so that it favors the flow”
13	Additional application experiment included	“Nonetheless, an application example described in the Supporting Information section 1 & 8 clearly

		demonstrates the superior capabilities for solution replacement of the existing diffusion cell setups, in particular for samples that cannot be flushed into the LC prior to the experiment.”
13	Reference added	 • Ref #35: E. Samson, J. Marchand & A. Snyder. Calculation of ionic diffusion coefficient on the basis of migration test results. Material and Structures 36, 156–165 (2003).
15	Wording improved	 • “The most obvious choice would be” • ”However, it was previously”
16	Recent Reference added	 • Fontana, M. et al. Electrochemical liquid phase TEM in aqueous electrolytes for energy applications: the role of liquid flow configuration. Preprint at https://www.researchsquare.com/article/rs-3660145/v1 (2023).
17	Statement on application examples added	“A series of model experiments were presented (here and in a follow-up manuscript) ³⁷ to demonstrate the drastic improvement of hydrodynamic conditions covering a broad range of applications.”
17	Funding Sources added	 • “KK-2023/00001” • “This study was carried out within the Ministerial Decree no. 1062/2021 and received funding from the FSE REACT-EU - PON Ricerca e Innovazione 2014-2020.”
18	Authors contribution updated	 • See manuscript for details
18	Competing Interest statement refined	“RIE etched chips were obtained from Protochips company in the frame of a collaborative project. These modified chips are not catalogue products but can be commercially requested from the company as of now. The authors declare no commercial interest in advertising this product.”
18	Wording improved, clarifying statement included	 • “laser sublimation” • “RIE etched chips were obtained from Protochips company in the frame of a collaborative project relying on standard deep RIE (DRIE, Bosch etch).³⁸”
18	Reference added	 • RIE etched chips were obtained from Protochips company in the frame of a collaborative project relying on standard deep RIE (DRIE, Bosch etch).³⁸ (Ref #38: Huff, M. Recent Advances in Reactive Ion Etching and Applications of High-Aspect-Ratio Microfabrication. Micromachines (Basel) 12, 991 (2021).)
20	Typo corrected	“height”

Supporting Information

Page #	Type of revision	Changes made
1	Authors List extended	 • Including Joscha Kruse,^{1,5} Maiara Aime Iriarte-Alonso,^{1,6} • Adding TECNIPESA IDENTIFICACION SL, Parque Empresarial Zuatzu, Edificio Donosti 1-3, 20018 Donostia-San Sebastián, Spain
2	Material list updated	 • “sodium citrate, sodium chloride, bis(p-sulfonatophenyl)-phenylphosphine, agarose and tetrachloroauric(III) acid were purchases from Sigma-Aldrich and used without any further purification.” • “All glassware was cleaned with aqua regia and rinsed thoroughly with Milli-Q water.”
3	Details on Application Example added	 • “1.4. Gold Nanoparticle (AuNP) Synthesis & Characterization [...]” • “1.5. Agarose Film Preparation & Characterization [...]” • “1.6. LP-TEM Imaging of Nanoscale Dynamics [...]”
3	Reference added	 • “Gold nanoparticles (AuNPs) were synthesized using the reversed Turkevich method for citrate-capped AuNPs.⁵” (Ref #5: Ojea-Jiménez, I., Bastús, N. G. & Puentes, V. Influence of the sequence of the reagents addition in the citrate-mediated synthesis of gold nanoparticles. Journal of Physical Chemistry C 115, 15752–15757 (2011).) • “Functionalization of AuNPs with bis(p-sulfonatophenyl)-phenylphosphine (BSPP) was described in an earlier work.⁶” (Ref #6: Kruse, J., Merkens, S., Chuvilin, A. & Grzelczak, M. Kinetic and Thermodynamic Hysteresis in Clustering of Gold Nanoparticles: Implications for Nanotransducers and Information Storage in Dynamic Systems. ACS Appl Nano Mater 3, 9520–9527 (2020).) • “To this end, manual scratching tests were performed as reported elsewhere^{7,8}” (Ref 7: Iriarte-Alonso, M. A., Bittner, A. M. & Chiantia, S. Influenza A virus hemagglutinin prevents extensive membrane damage upon dehydration. BBA Advances 2, 100048 (2022).; Ref #8: Üzümlü, C., Hellwig, J., Madaboosi, N., Volodkin, D. & von Klitzing, R. Growth behaviour and mechanical

		properties of PLL/HA multilayer films studied by AFM. Beilstein Journal of Nanotechnology 3, 778–788 (2012).)
6	Outline improved	“2.5. Flow Channel Resistance in LP-TEM Flow Systems with Diffusion Cells “
8	Subheader updated	“3.1. In-depth discussion of virtual Prototyping (Fig. 3)“
8	Abbreviation introduced	“the imaging area (IA)”
9	Additional Information provided	“3.2. Limitations of Diffusion Cell Design [...]”
10	Supporting Information restructured	“4. Mass Transport Mechanisms in Diffusion Cell Setups ”
10	Additional Information provided	“To evaluate the dominant mass transport [...]”
12, 13, 14	Subheader updated, additional information provided, additional data provided, additional model provided	 “4.2. Effect of Window Bulging [...]”
18, 20	Referencing updated	 “Fig. S14 and S15” “Fig. S16d”
19	Typo corrected	“ KMnO₄ solution”
22	Reference added	 Ref #12: E. Samson, J. Marchand & A. Snyder. Calculation of ionic diffusion coefficient on the basis of migration test results. Material and Structures 36, 156–165 (2003).
24	Application Example included	“7. Application Experiment [...]”
25	Reference added	 “While the hopping indicates increased particle motion due to high ionic strength,¹¹” (Ref #13: Woehl, T. J. & Prozorov, T. The Mechanisms for Nanoparticle Surface Diffusion and Chain Self-Assembly Determined from Real-Time Nanoscale Kinetics in Liquid. Journal of Physical Chemistry C 119, 21261–21269 (2015).) In fact, radiolytic damage of bio-polymeric composite materials has been widely observed in similar scenarios.^{12,13} (Ref #14: Korpanty, J., Parent, L. R. & Gianneschi, N. C. Enhancing and Mitigating Radiolytic Damage to Soft Matter in Aqueous Phase Liquid-Cell Transmission Electron Microscopy in the Presence of Gold Nanoparticle Sensitizers or Isopropanol Scavengers. Nano Lett 21, 1141–1149

		(2021).; Ref #15: Parent, L. R., Gnanasekaran, K., Korpany, J. & Gianneschi, N. C. 100th Anniversary of Macromolecular Science Viewpoint: Polymeric Materials by In Situ Liquid-Phase Transmission Electron Microscopy. ACS Macro Lett 10, 14–38 (2021).) • Accounting for hindrance of diffusion in agarose gel,¹⁴ (Ref #16: Johnson, E. M., Berk, D. A., Jain, R. K. & Deen, W. M. Hindered Diffusion in Agarose Gels: Test of Effective Medium Model. Biophys J 70, 1017–1026 (1996).)
--	--	---

REVIEWERS' COMMENTS

Reviewer #1 (Remarks to the Author):

Apologies for my delayed response.

I am pleased with the revisions made to the manuscript and the inclusion of an application example. I believe the manuscript is now suitable for publication. The modifications to the chip should enhance its imaging capabilities and facilitate further research.

Reviewer #2 (Remarks to the Author):

Reviewer #3 (Remarks to the Author):

The authors have done a great job addressing the reviewers' comments and revising the documents accordingly. I still recommend being cautious about (and maybe considering adjusting) superlative wording (e.g., "the fastest" or "the optimal design" etc.) because the article is still outstanding without such statements. In all, I recommend publishing this article soon, so that the obtained knowledge and insights can be spread to interested readers. Congratulations to the authors on this great work.

Reviewer #4 (Remarks to the Author):

The authors have significantly improved the article and have clarified the pressing queries that were mentioned in the previous review. The authors have presented a new TEM liquid flow cell with improved mixing time constant. This publication advances the field by providing a platform for furthering time-resolved studies with TEM. I would suggest the manuscript to be published as it is.

Reviewer #5 (Remarks to the Author):

I'm fine with the author's response, they addressed all of my comments appropriately. I can therefore recommend their manuscript for publication in Nature Communications.

Reviewer #6 (Remarks to the Author):

The authors have addressed all concerns with the manuscript.

REVIEWERS' COMMENTS

Reviewer #1 (Remarks to the Author):

Apologies for my delayed response.

I am pleased with the revisions made to the manuscript and the inclusion of an application example. I believe the manuscript is now suitable for publication. The modifications to the chip should enhance its imaging capabilities and facilitate further research.

Reviewer #2 (Remarks to the Author):

Reviewer #3 (Remarks to the Author):

The authors have done a great job addressing the reviewers' comments and revising the documents accordingly. I still recommend being cautious about (and maybe considering adjusting) superlative wording (e.g., "the fastest" or "the optimal design" etc.) because the article is still outstanding without such statements. In all, I recommend publishing this article soon, so that the obtained knowledge and insights can be spread to interested readers. Congratulations to the authors on this great work.

Reviewer #4 (Remarks to the Author):

The authors have significantly improved the article and have clarified the pressing queries that were mentioned in the previous review. The authors have presented a new TEM liquid flow cell with improved mixing time constant. This publication advances the field by providing a platform for furthering time-resolved studies with TEM. I would suggest the manuscript to be published as it is.

Reviewer #5 (Remarks to the Author):

I'm fine with the author's response, they addressed all of my comments appropriately. I can therefore recommend their manuscript for publication in Nature Communications.

Reviewer #6 (Remarks to the Author):

The authors have addressed all concerns with the manuscript.

We are pleased to have been able to entirely satisfy the reviewers' concerns. We thank all reviewers for their comments which have allowed us to substantially improve the quality of the manuscript making it suited for Nature Communications!